# EnvSocial-Diff: A Diffusion-Based Crowd Simulation Model with Environmental Conditioning and Individual-Group Interaction

**Bingxue Zhao**    **Qi Zhang**[*]    **Hui Huang**
VCC, College of Computer Science and Software Engineering, Shenzhen University
`2410103030@mails.szu.edu.cn, qi.zhang.opt@gmail.com,`
`hhzhiyan@gmail.com`

## ABSTRACT

Modeling realistic pedestrian trajectories requires accounting for both social interactions and environmental context, yet most existing approaches largely emphasize social dynamics. We propose **EnvSocial-Diff**: a diffusion-based crowd simulation model informed by social physics and augmented with environmental conditioning and individual–group interaction. Our structured environmental conditioning module explicitly encodes obstacles, objects of interest, and lighting levels, providing interpretable signals that capture scene constraints and attractors. In parallel, the individual–group interaction module goes beyond individual-level modeling by capturing both fine-grained interpersonal relations and group-level conformity through a graph-based design. Experiments on multiple benchmark datasets demonstrate that EnvSocial-Diff outperforms the latest state-of-the-art methods, underscoring the importance of explicit environmental conditioning and multi-level social interaction for realistic crowd simulation.

## 1 INTRODUCTION

Crowd simulation plays an important role in modeling and predicting the collective behavior of pedestrians in dynamic environments, with applications ranging from virtual reality and digital twin systems to public safety management and urban planning (Musse et al., 2021). A central goal is to generate realistic walking trajectories for multiple agents while accounting for social interactions and environmental constraints. Over the years, numerous methods have been proposed to address this task, ranging from rule-based approaches (Reynolds, 1987) and force-based models (Helbing & Molnar, 1995; Kolivand et al., 2021; Chraibi et al., 2011) to data-driven learning-based approaches (Alahi et al., 2016; Gupta et al., 2018; Lee et al., 2018; Charalambous et al., 2023). Among them, the Social Force Model (SFM) (Helbing & Molnar, 1995) and its extensions have been widely adopted for their

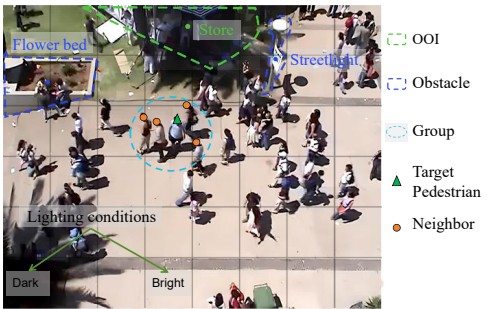

Figure 1: Environmental factors are important in crowd simulation. The target pedestrian is influenced by nearby neighbors, obstacles, objects of interest (OOI), and lighting conditions. The scene image is divided into grids to calculate lighting information.

interpretable and physically grounded structure. More recently, physics-informed generative approaches have emerged. In particular, the Social Physics Informed Diffusion Model (SPDiff) (Chen et al., 2024) integrates a conditional diffusion process into the Social Force Model, where the diffusion module refines predicted accelerations based on historical motion and individual-level social interactions. While SFM inherently accounts for basic obstacle avoidance through repulsive forces, SPDiff does not explicitly incorporate structured environmental conditioning (e.g., objects of in-

---

[*]Corresponding author.

terest or lighting), nor does it capture group-level conformity, leaving important behavioral factors underexplored.

A core challenge in crowd simulation lies in capturing the diverse factors that shape pedestrian behavior, including social interactions such as collision avoidance, group coherence, and route choice. While most existing approaches (Mohamed et al., 2020; Xu et al., 2022; Kim et al., 2024; Itatani & Pelechano, 2024; Pascoli et al., 2025) have primarily focused on modeling these interactions through graph-based, recurrent, or probabilistic frameworks, they typically exhibit two major limitations. First, social modeling is often restricted to individual-level interactions, overlooking higher-level group conformity that strongly influences collective motion. Second, the treatment of the environment is oversimplified: most methods, including physics-informed models such as SPDiff, primarily account for obstacles through repulsive forces or binary traversable maps, but do not explicitly encode richer contextual cues. This abstraction neglects influences such as attractive objects of interest (OOI) (e.g., stores, kiosks), which act as behavioral attractors guiding route choice (Tong & Bode, 2022), and perceptual cues like lighting (as illustrated in Figure 1), which have been shown in psychology and urban design studies to affect safety perception, comfort, and walking preferences (Warren et al., 2001; Hao et al., 2022; Liu et al., 2022). Addressing these gaps motivates the need for a unified framework that explicitly integrates structured environmental conditioning with multi-level social modeling.

To overcome these limitations, we propose **EnvSocial-Diff** (see Figure 2), a social physics-informed diffusion model that jointly models structured environmental conditioning and individual-group interaction. On the environment side, we encode obstacles, objects of interest (OOI), and lighting as conditional signals that guide the generative denoising process. On the social side, an Individual–Group Interaction module captures both individual-level relations and group-level conformity via a graph-based design. These two pillars are fused with historical trajectories and a destination attraction term, forming four complementary components that jointly condition the diffusion model to produce socially compliant, context-aware, and realistic trajectory predictions.

In summary, the paper's contributions are as follows.

- We propose **EnvSocial-Diff**, a diffusion-based crowd simulation model informed by social physics, which unifies structured environmental conditioning with social interaction modeling.
- We design structured environmental encoders that explicitly model obstacles, objects of interest, and lighting, and integrate them with an Individual–Group Interaction (IGI) module that captures both fine-grained interpersonal relations and group-level conformity. This unified design enables physically interpretable trajectory predictions.
- Experiments on GC and UCY benchmarks show that EnvSocial-Diff outperforms state-of-the-art baselines across multiple trajectory metrics, validating the effectiveness of explicit environmental conditioning and multi-level social interaction modeling.

## 2    RELATED WORK

**Physics-based Crowd Simulation.**    Early approaches relied on handcrafted rules and physics-inspired models. The Boids model (Reynolds, 1987) simulated collective animal motion through simple rules of separation, alignment, and cohesion. A milestone, the Social Force Model (SFM) (Helbing & Molnar, 1995), introduced psychological forces such as goal attraction and social repulsion, enabling realistic simulation of pedestrian interactions. Other approaches include Cellular Automata (CA) (Sarmady et al., 2010), which discretize time and space for efficient simulation but lack motion continuity, and Velocity Obstacle (VO) methods (Fiorini & Shiller, 1998) and their variants (RVO (Van den Berg et al., 2008), ORCA (Snape et al., 2010), HRVO (Van Den Berg et al., 2011)), which use geometric constraints for collision avoidance. Inspired by fluid dynamics, continuum-based models (Hughes, 2002; Huang et al., 2009; Liang & Du, 2021) treat crowds as continuous media, capturing macroscopic flow in dense settings. While these methods laid foundational groundwork, their reliance on predefined rules and physical simplifications limits their ability to model complex, context-aware human behaviors.

**Data-driven Social Modeling.**    Recent methods leverage learning-based frameworks to capture pedestrian interactions. Early works such as Social LSTM (Alahi et al., 2016) and Social GAN

Table 1: Summary of main notations used in EnvSocial-Diff.

| Symbol | Meaning | Symbol | Meaning |
|---|---|---|---|
| $i, j$ | Pedestrian indices | $t$ | Time step index |
| $S_i^t$ | State of $i$ at $t$, $[\vec{p}_i^t, \vec{v}_i^t, \vec{a}_i^t]$ | $H$ | Prediction horizon (number of future steps) |
| $\vec{p}_i^t$ | Position of $i$ at time $t$ | $\vec{v}_i^t$ | Velocity of $i$ at time $t$ |
| $\vec{a}_i^t$ | Acceleration of $i$ at time $t$ | $f_\theta$ | Denoising network in acceleration space |
| $\vec{F}_i^{\text{dest}}$ | Destination-driving force of $i$ | $c_i^t$ | Conditioning signals $[\vec{F}_i^{\text{env}} \oplus \vec{F}_i^{\text{social}} \oplus \vec{F}_i^{\text{hist}}]$ |
| $\vec{F}_i^{\text{env}}$ | Environment-induced force on $i$ | $\vec{F}_i^{\text{social}}$ | Social interaction force on $i$ |
| $\vec{F}_i^{\text{hist}}$ | History-based force on $i$ | $k$ | Diffusion step index $(1, \ldots, K)$ |
| $\mathbf{y}_{i,k}$ | Noisy acceleration at step $k$ | $\mathbf{y}_{i,0}^t$ | Clean target acceleration of $i$ at time $t$ |
| $\mathcal{O}$ | Set of obstacles | $\mathcal{I}$ | Set of objects of interest |
| $\mathcal{L}$ | Set of lighting cells | $\mathcal{M}$ | Environment entities, $\mathcal{M} = \mathcal{O} \cup \mathcal{I} \cup \mathcal{L}$ |
| $f^{sc}$ | Global scene feature | $f_{\text{light}}^{\text{raw}}$ | Spatial lighting vector |
| $f_l^{\text{obs}}$ | Feature of the $l$-th obstacle | $f_m^{\text{ooi}}$ | Feature of the $m$-th object-of-interest (OOI) |
| $\text{sim}_{ij}^1$ | Approach tendency | $\text{sim}_{ij}^2$ | Motion alignment |
| $\text{sim}_i^3$ | Group conformity | $r_{ij}$ | Relative motion descriptor $[\Delta\vec{p}_{ij} \oplus \Delta\vec{v}_{ij}]$ |

(Gupta et al., 2018) employed recurrent and generative models for individual-level interaction modeling, while STGCNN (Mohamed et al., 2020) introduced spatio-temporal graphs for relational reasoning. Subsequent approaches, including SocialCircle (Wong et al., 2024), HighGraph (Kim et al., 2024), and RSBS (Sun et al., 2020), enhance modularity and temporal dynamics. Other directions explore interpretable latent modeling (SocialVAE (Xu et al., 2022)), endpoint conditioning (PECNet (Mangalam et al., 2020)), and multimodal diffusion-based prediction (MID (Gu et al., 2022)).

In parallel, some methods attempt to integrate scene context. Scene-aware approaches leverage semantic maps or global embeddings (Manh & Alaghband, 2018; Mangalam et al., 2021; Ngiam et al., 2022; Bae et al., 2025; Yuan et al., 2021), which provide high-level semantic awareness but lack behavioral modeling. NSP (Yue et al., 2022), which we also include as a baseline in our experiments, introduces a physics-inspired framework that fuses social interactions with environmental cues. In NSP, the environment is represented by segmenting scenes into walkable and non-walkable regions, where non-walkable areas in a pedestrian's field of view exert repulsive forces. However, this binary abstraction overlooks richer environmental roles such as attractive objects of interest (OOI) or perceptual cues like lighting, and its social modeling remains limited to the individual level. UniTraj (Feng et al., 2024) further proposes a unified environmental network for short-horizon trajectory prediction, but it similarly relies on global semantic context rather than structured environmental conditioning. Overall, these environment-aware approaches are confined to short-term forecasting and do not integrate multi-level social modeling, leaving factors underexplored.

**Physics-informed Generative Approaches.** To improve long-term prediction, physics-informed generative methods combine physical priors with data-driven learning. PCS (Zhang et al., 2022) integrates physical constraints with trajectory forecasting, while SPDiff (Chen et al., 2024) introduces a diffusion model conditioned on social forces derived from the Social Force Model (SFM). In SPDiff, the diffusion process refines accelerations based on historical motion and individual-level interactions. While the SFM formulation inherently includes obstacle avoidance, SPDiff does not explicitly incorporate structured environmental conditioning (e.g., OOI, lighting) or group-level conformity, leaving important behavioral influences underexplored.

*In contrast, our work introduces structured environmental conditioning—decomposing the environment into obstacles, OOI, and lighting—and complements it with an Individual–Group Interactions module, enabling unified modeling of environmental and social influences for realistic long-horizon crowd simulation.*

## 3 METHOD

We propose **EnvSocial-Diff** (see Figure 2), a diffusion-based crowd simulation model informed by social physics. Following SPDiff (Chen et al., 2024), the destination attraction force is applied outside the diffusion process, thereby preserving long-term intent, while historical trajectories are

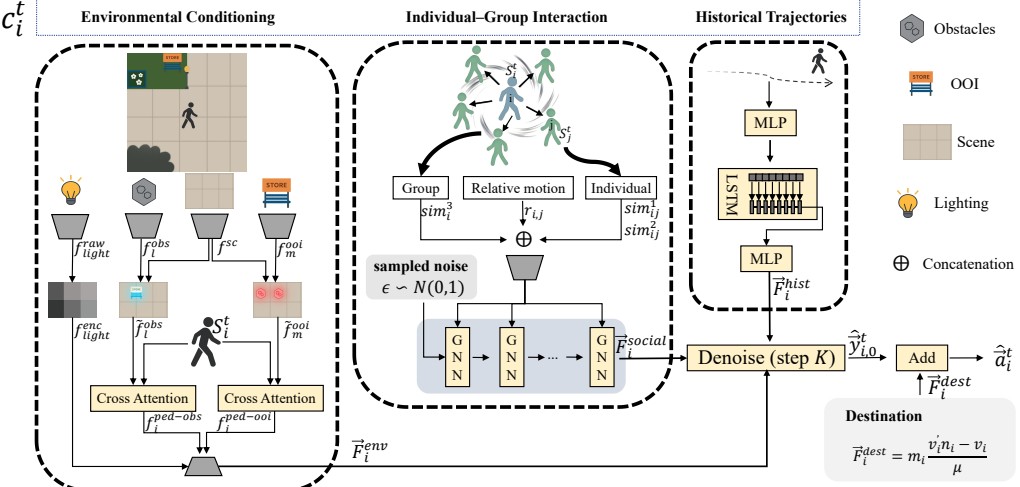

Figure 2: EnvSocial-Diff pipeline. Pedestrian motion is modeled as in the Social Force Model (SFM), where the destination force $\vec{F}_i^{\text{dest}}$ is applied outside the diffusion process to preserve long-term intent. The conditioning signals $c_i^t = [\vec{F}_i^{\text{env}} \oplus \vec{F}_i^{\text{social}} \oplus \vec{F}_i^{\text{hist}}]$ aggregate three interactive components: (1) **Environmental Conditioning** — obstacle and OOI features are encoded via cross-attention with pedestrians, while lighting features are extracted from grid-based scene brightness; (2) **Individual–Group Interactions** — GNNs encode individual-level ($\text{sim}_{ij}^1$, $\text{sim}_{ij}^2$), group-level ($\text{sim}_i^3$), and relative motion ($r_{ij}$) to produce the social force $\vec{F}_i^{\text{social}}$; and (3) **Historical Trajectories** — short-term motion trends are encoded from recent states using an LSTM. Given $c_i^t$ and Gaussian noise $\epsilon \sim \mathcal{N}(0,1)$, the denoiser $f_\theta$ performs reverse diffusion to recover clean accelerations $\hat{\mathbf{y}}_{i,0}^t$, which are then combined with the destination force to yield the final prediction $\hat{\vec{a}}_i^t$.

encoded via a unidirectional LSTM. **Unlike SPDiff, which incorporates environmental conditioning only implicitly via the classical Social Force Model (SFM) formulation, we explicitly model structured environmental factors through obstacles, objects of interest, and lighting.** Furthermore, we introduce an *Individual–Group Interaction (IGI)* module that captures both individual-level relations and group-level conformity. Together, these components—environment, IGI, and history—constitute the conditioning signals $c_i^t$ to the diffusion model, while the destination force is injected during rollout. This design enables trajectory forecasts that are context-aware, socially compliant, and physically interpretable.

The model takes as input the 2D start and destination coordinates of each pedestrian, together with the scene environment information. The environment consists of the BEV image and its textual description, together with structured environment entities $\mathcal{M}$ (summarized in Table 1). For obstacles $\mathcal{O}$ and objects of interest $\mathcal{I}$, each entity is represented by its cropped image patch, textual description, and 2D position in the BEV frame. For the lighting factor $\mathcal{L}$, we extract the V-channel from the BEV image in HSV space to form a global lighting map, which serves as the lighting input. Given these inputs, the model generates full trajectories for all pedestrians by predicting their accelerations over time.

## 3.1 OVERALL MODEL

As shown in Eq. (1), pedestrian motion follows the SFM (Helbing & Molnar, 1995), where multiple interactive forces jointly govern pedestrian dynamics:

$$\vec{F}_i = \vec{F}_i^{\text{dest}} + \underbrace{\left( \vec{F}_i^{\text{hist}} + \sum_{j \in \mathcal{N}eigh_i} \vec{F}_{ij}^{\text{social}} + \sum_{h \in \mathcal{M}} \vec{F}_{ih}^{\text{env}} \right)}_{\text{conditioning signals}}. \tag{1}$$

Here, $\vec{F}_i^{\text{dest}}$ drives pedestrian $i$ toward its destination and is defined following SFM as $\vec{F}_i^{\text{dest}} = m_i \frac{v_i' n_i - v_i}{\mu}$, where $v_i$ is the current velocity, $v_i'$ is the desired walking speed, and $n_i$ is the direction

towards the destination. $m_i$ is a coefficient for individuals while $\mu$ is a global coefficient. Unlike the other forces, this term is applied outside the diffusion process during rollout to preserve long-term intent. The remaining terms constitute the *conditioning signals*: $\vec{F}_i^{\text{hist}}$ encodes short-term motion trends from recent trajectories, $\sum_{j \in \mathcal{N}eigh_i} \vec{F}_{ij}^{\text{social}}$ models individual–group interactions, and $\sum_{h \in \mathcal{M}} \vec{F}_{ih}^{\text{env}}$ incorporates structured scene context ($\mathcal{M} = \mathcal{O} \cup \mathcal{I} \cup \mathcal{L}$ for obstacles, objects of interest, and lighting).

As acceleration is proportional to the net force ($\vec{F} = m\vec{a}$), we predict future accelerations rather than positions. This yields a physically grounded representation of motion dynamics. To model this process, we employ a diffusion model (Ho et al., 2020) within the acceleration space. Specifically, the conditioning signals in Eq. (1) are modeled by the denoiser output. The destination force is then added outside the diffusion process to obtain the final acceleration. The state of pedestrian $i$ at time $t$ is denoted as $S_i^t = [\vec{p}_i^t, \vec{v}_i^t, \vec{a}_i^t]$ denote position, velocity, and acceleration, respectively. Our goal is to generate the sequence of predicted accelerations $\{\hat{\vec{a}}_i^{t+1}, \ldots, \hat{\vec{a}}_i^{t+H}\}$ over the prediction horizon $H$.

In the forward process, we progressively add Gaussian noise to the ground-truth accelerations $\mathbf{y}_{i,0}^t$, forming a Markov chain that transforms the clean accelerations into approximately pure noise:

$$q(\mathbf{y}_{i,k} \mid \mathbf{y}_{i,k-1}) = \mathcal{N}\left(\sqrt{1 - \beta_k}\, \mathbf{y}_{i,k-1},\, \beta_k \mathbf{I}\right), \tag{2}$$

where $\beta_k$ denotes the variance at step $k$ in the noise schedule, with $k = 1, \ldots, K$.

During inference, the reverse process starts from Gaussian noise and iteratively denoises it into accelerations conditioned on $c_i^t = [\vec{F}_i^{\text{env}} \oplus \vec{F}_i^{\text{social}} \oplus \vec{F}_i^{\text{hist}}]$. A neural network $f_\theta$ parameterizes this process by predicting clean accelerations at each step conditioned on $c_i^t$.

$$p_\theta(\mathbf{y}_{i,k-1} \mid \mathbf{y}_{i,k}, c_i^t) = q\left(\mathbf{y}_{i,k-1} \mid \mathbf{y}_{i,k}, f_\theta(\mathbf{y}_{i,k}, k, c_i^t)\right). \tag{3}$$

To encode $\vec{F}_i^{\text{hist}}$, we use the recent trajectory sequence $\{S_i^{t-L+1}, \ldots, S_i^t\}$ over an observation window of length $L$. The sequence is passed through linear projections and a unidirectional LSTM encoder, and the final hidden state provides a compact temporal feature for conditioning the denoiser (see Figure 2, Historical Trajectories block).

After obtaining the denoised conditioning accelerations $\hat{\mathbf{y}}_{i,0}^t$, we add the destination force $\vec{F}_i^{\text{dest}}$ to produce the final predicted accelerations $\hat{\vec{a}}_i^t$. These accelerations are then used to recursively update the velocity and position of each pedestrian via standard kinematic equations.

$$\vec{v}_i^{t+\tau} = \vec{v}_i^{t+\tau-1} + \hat{\vec{a}}_i^{t+\tau}\, \Delta t, \tag{4}$$

$$\vec{p}_i^{t+\tau} = \vec{p}_i^{t+\tau-1} + \vec{v}_i^{t+\tau-1}\, \Delta t + \tfrac{1}{2}\hat{\vec{a}}_i^{t+\tau}\, \Delta t^2, \tag{5}$$

where $\tau = 1, \ldots, H$ and $\Delta t$ denotes the time step.

## 3.2 STRUCTURED ENVIRONMENTAL CONDITIONING

The environmental conditioning module explicitly encodes structured scene elements—including obstacles, objects of interest (OOI, e.g., stores, kiosks, benches), and lighting—as in the *Environmental Conditioning* block of Figure 2. Obstacle and OOI features are first enhanced with global scene context, after which they interact with pedestrians via cross-attention, while lighting features are extracted from grid-based brightness in the HSV space. Together, these features provide repulsive, attractive, and contextual cues that influence pedestrian motion.

For obstacles and objects of interest (OOI), GPT is first used to produce concise textual descriptions. The scene-level BEV image and the cropped image patches are encoded using ResNet-50 to obtain visual embeddings, while the textual descriptions are encoded using BERT. The resulting visual and textual embeddings are then concatenated and projected to respectively form the global scene feature $f^{sc}$, the obstacle features $f_l^{\text{obs}}$, and the OOI features $f_m^{\text{ooi}}$.

**Obstacles.** We model obstacle influence in two stages of cross-attention. First, each obstacle feature $f_l^{\text{obs}}$ is enhanced using the global scene feature $f^{sc}$:

$$Q_l^{\text{obs}} = \text{Proj}_Q(f_l^{\text{obs}} \oplus p_l^{\text{obs}}), \quad K^{sc} = \text{Proj}_K(f^{sc}), \quad V^{sc} = \text{Proj}_V(f^{sc}), \tag{6}$$

$$\tilde{f}_l^{\text{obs}} = \text{Attention}(Q_l^{\text{obs}}, K^{sc}, V^{sc}), \tag{7}$$

where $p_l^{\text{obs}}$ is the obstacle position. This produces context-enhanced obstacle features $\tilde{f}_l^{\text{obs}}$.

Second, pedestrians attend to the enhanced obstacle features to capture obstacle–pedestrian interactions:

$$f_i^{\text{ped-obs}} = \sum_{l \in \mathcal{O}} \text{softmax}_l \left( \frac{Q_i^\top K_l^{\text{obs}}}{\sqrt{d_1}} + b(\vec{p}_{i,l}^{\text{rel}}) \right) V_l^{\text{obs}}, \tag{8}$$

where $Q_i = W_Q S_i^t$ encodes pedestrian $i$'s state $S_i^t = [\vec{p}_i^t, \vec{v}_i^t, \vec{a}_i^t]$, $K_l^{\text{obs}} = W_K \tilde{f}_l^{\text{obs}}$, $V_l^{\text{obs}} = W_V \tilde{f}_l^{\text{obs}}$, $\vec{p}_{i,l}^{\text{rel}} = \vec{p}_l^{\text{obs}} - \vec{p}_i^t$ is the relative position, $b(\cdot)$ is a small neural network, and $d_1$ is the feature dimensionality for scaling.

**Objects of Interest (OOI).** OOI serves as a semantic attractor that influences route choice. Unlike obstacles, which require fine-grained avoidance behavior, OOI primarily provides global semantic cues. Therefore, each OOI feature $f_m^{\text{ooi}}$ is enhanced by concatenating its positional encoding $p_m^{\text{ooi}}$ and the global scene feature $f^{sc}$, followed by a projection:

$$\tilde{f}_m^{\text{ooi}} = \text{Proj}(f_m^{\text{ooi}} \oplus p_m^{\text{ooi}} \oplus f^{sc}). \tag{9}$$

The interaction with pedestrians is then modeled via cross-attention:

$$f_i^{\text{ped-ooi}} = \sum_{m \in \mathcal{I}} \text{softmax}_m \left( \frac{Q_i^\top K_m^{\text{ooi}}}{\sqrt{d_2}} + b(\vec{p}_{i,m}^{\text{rel}}) \right) V_m^{\text{ooi}}, \tag{10}$$

where $Q_i = W_Q S_i^t$, $K_m^{\text{ooi}} = W_K \tilde{f}_m^{\text{ooi}}$, $V_m^{\text{ooi}} = W_V \tilde{f}_m^{\text{ooi}}$, $\vec{p}_{i,m}^{\text{rel}} = \vec{p}_m^{\text{ooi}} - \vec{p}_i^t$, $b(\cdot)$ is a small neural network, and $d_2$ is the feature dimensionality for scaling.

**Lighting.** We treat lighting as a global contextual factor rather than localized entities. The scene image is divided into grids, and the average value of the V-channel in HSV space within each cell is pooled to form a spatial lighting vector $\mathbf{f}_{\text{light}}^{\text{raw}}$, which is then encoded by a lightweight MLP, resulting into the lighting feature $f_{\text{light}}^{\text{enc}}$:

$$f_{\text{light}}^{\text{enc}} = \text{MLP}(f_{\text{light}}^{\text{raw}}). \tag{11}$$

This design is supported by prior psychophysical evidence showing that lighting strongly influences pedestrian movement (Rahm & Johansson, 2018), which reports that outdoor lighting improves walkability and facilitates obstacle detection.

**Pedestrian–Environment Feature Aggregation.** Finally, the influences from obstacles, OOI, and lighting are aggregated into a unified environment-aware feature for pedestrian $i$ by concatenation followed by an MLP:

$$\vec{F}_i^{\text{env}} = \text{MLP}\Big( f_i^{\text{ped-obs}} \oplus f_i^{\text{ped-ooi}} \oplus f_{\text{light}}^{\text{enc}} \Big). \tag{12}$$

### 3.3 Individual–Group Interaction (IGI)

The IGI module encodes social influences at two levels, as illustrated in the *IGI* block of Figure 2. At the individual level, similarity measures capture approach tendency and motion alignment between pedestrian $i$ and its neighbors $j \in \mathcal{N}eigh_i$. At the group level, a conformity measure models the alignment of $i$ with the surrounding group conformity. In addition, relative motion descriptors $r_{ij} = \Delta \vec{p}_{ij} \oplus \Delta \vec{v}_{ij}$ provide complementary spatial and velocity cues. These descriptors are aggregated by a multi-layer GNN to produce the social force feature $\vec{F}_i^{\text{social}}$, which serves as part of the conditioning input.

**Individual-level similarities.** We introduce two measures to capture pairwise relations between pedestrian $i$ and neighbor $j$:

- **Approach tendency** $\text{sim}_{ij}^1$ quantifies whether $j$ is moving toward $i$, reflecting potential collision risk:

$$\text{sim}_{ij}^1 = \frac{1}{2} \left( \frac{\Delta \vec{p}_{ij}}{\|\Delta \vec{p}_{ij}\|} \cdot \frac{\vec{v}_j}{\|\vec{v}_j\|} + 1 \right), \tag{13}$$

where $\Delta \vec{p}_{ij} = \vec{p}_j - \vec{p}_i$. This is the cosine similarity between the normalized relative position and the neighbor's velocity, mapped to $[0, 1]$. Larger values indicate that $j$ is approaching $i$ more directly.

- **Motion alignment** $\text{sim}_{ij}^2$ measures the directional consistency of their velocities:

$$\text{sim}_{ij}^2 = \tfrac{1}{2}\left(\frac{\vec{v}_i}{\|\vec{v}_i\|}\cdot\frac{\vec{v}_j}{\|\vec{v}_j\|}+1\right). \tag{14}$$

Higher values indicate stronger velocity alignment.

**Group-level similarity.** Beyond individual relations, pedestrians are influenced by the collective motion of surrounding neighbors. We define a **group conformity** similarity $\text{sim}_i^3$ by comparing the motion state of pedestrian $i$ with the neighborhood average:

$$\text{sim}_i^3 = \tfrac{1}{2}\left(\frac{w_i}{\|w_i\|}\cdot\frac{g_i}{\|g_i\|}+1\right), \tag{15}$$

where $w_i = \vec{v}_i \oplus \vec{a}_i$ encodes the velocity and acceleration of $i$, and $g_i = \frac{1}{|\mathcal{N}eigh_i|}\sum_{j\in\mathcal{N}eigh_i}(\vec{v}_j \oplus \vec{a}_j)$ denotes the average motion of its neighbors, and $|\mathcal{N}eigh_i|$ is the number of neighbors. This similarity, normalized to $[0,1]$, reflects the degree to which pedestrian $i$ conforms to the surrounding group dynamics; larger values indicate stronger conformity.

**GNN Aggregation.** To instantiate the social force $\vec{F}_i^{\text{social}}$ in Eq. (1), we employ a multi-layer graph neural network (GNN) that aggregates the individual-level and group-level similarities defined above. Each pedestrian $i$ is represented as a node, initialized as:

$$h_i^0 = \text{MLP}_{\text{init}}(S_i^t \oplus \boldsymbol{\epsilon}_i^t \oplus g_i), \tag{16}$$

where $S_i^t = [\vec{p}_i^t, \vec{v}_i^t, \vec{a}_i^t]$ is the pedestrian's state, $\boldsymbol{\epsilon}_i^t$ denotes sampled noise, and $g_i$ encodes the neighborhood average motion. At each GNN layer $l_g$, the edge feature between pedestrian $i$ and neighbor $j$ incorporates relative motion and the similarity measures:

$$e_{ij} = r_{ij} \oplus \text{sim}_{ij}^1 \oplus \text{sim}_{ij}^2 \oplus \text{sim}_i^3, \tag{17}$$

where $r_{ij}$ denotes the relative motion descriptor. These edge features capture spatial proximity, motion cues, and social affinity, and are transformed by a shared edge-level multilayer perceptron $\text{MLP}_{\text{edge}}$ to generate messages.

Each node updates its representation by concatenating its current hidden state $h_i^{l_g}$, the mean-aggregated messages from neighbors, and the normalized local group feature $N_g^i = \text{Norm}(g_i)$, followed by a node-level transformation:

$$h_i^{l_g+1} = \text{MLP}_{\text{node}}\left(h_i^{l_g}\oplus\frac{1}{|\mathcal{N}eigh_i|}\sum_{j\in\mathcal{N}eigh_i}\text{MLP}_{\text{edge}}(e_{ij})\oplus N_g^i\right). \tag{18}$$

This updated hidden state is progressively refined through $L_g$ layers within the IGI module. Finally, a task-specific output MLP predicts the social interaction force for pedestrian $i$:

$$\vec{F}_i^{\text{social}} = \text{MLP}_{\text{out}}(h_i^{L_g}). \tag{19}$$

### 3.4 Denoising and Multi-Frame Rollout Training

After reverse diffusion, the denoised conditioning acceleration $\hat{y}_{i,0}^t$ is combined with the destination term to obtain the final acceleration $\hat{\vec{a}}_i^t$. We train the model under a multi-frame rollout strategy (Chen et al., 2024), and optimize a weighted mean-squared error over accelerations and positions:

$$\mathcal{L} = \frac{1}{NH}\sum_{i=1}^{N}\sum_{\tau=1}^{H}\left(\lambda_a\|\hat{\vec{a}}_i^{t+\tau} - \vec{a}_i^{t+\tau}\|_2^2 + \lambda_p\|\hat{\vec{p}}_i^{t+\tau} - \vec{p}_i^{t+\tau}\|_2^2\right), \tag{20}$$

where $\vec{a}_i^{t+\tau}, \vec{p}_i^{t+\tau}$ are ground truth, $\hat{\vec{a}}_i^{t+\tau}, \hat{\vec{p}}_i^{t+\tau}$ are predictions, $N$ is the number of pedestrians, and $\lambda_a, \lambda_p$ are loss weights.

Table 2: Quantitative comparison on GC and UCY datasets. Results for baselines are directly reported from SPDiff (Chen et al., 2024), except for E-$V^2$-SC and **Ours**, which are reproduced under the same experimental settings. Here, **Ours** corresponds to our proposed **EnvSocial-Diff** model.

| Group | Models | GC | | | | | | UCY | | | | | |
|---|---|---|---|---|---|---|---|---|---|---|---|---|---|
| | | MAE↓ | OT↓ | FDE↓ | MMD↓ | DTW↓ | Col↓ | MAE↓ | OT↓ | FDE↓ | MMD↓ | DTW↓ | Col↓ |
| Physics-based | CA | 2.7080 | 5.4990 | - | 0.0620 | - | 1492 | 8.3360 | 79.4200 | - | 2.0220 | - | 4504 |
| | SFM | 1.2590 | 2.1140 | - | 0.0150 | - | **622** | 2.5390 | 6.5710 | - | 0.1290 | - | 434 |
| Data-driven | STGCNN | 8.1608 | 15.8372 | - | 0.5296 | 5.1438 | 2076 | 7.5121 | 18.7721 | - | 0.5149 | 5.1695 | 1348 |
| | PECNet | 2.0669 | 4.3054 | - | 0.0397 | 0.7431 | 1142 | 3.9674 | 16.1412 | - | 0.1504 | 2.0986 | 1348 |
| | MID | 8.4257 | 35.1797 | - | 0.3737 | 4.2773 | 1620 | 8.2915 | 47.8711 | - | 0.4384 | 4.7109 | 1076 |
| | E-$V^2$-SC | 8.8816 | 52.5596 | 7.2464 | 1.8844 | 8.8851 | >9999 | 8.8591 | 60.5391 | 9.5011 | 1.1427 | 8.8972 | >9999 |
| Physics-informed | PCS | 1.0320 | 1.5963 | - | 0.0126 | 0.4378 | 764 | 2.3134 | 6.2336 | - | 0.1070 | 0.9887 | **238** |
| | NSP | 0.9884 | 1.4893 | - | 0.0106 | 0.3329 | 734 | 2.4006 | 6.3795 | - | 0.1199 | 0.9965 | 380 |
| | SPDiff | 0.9116 | 1.3925 | - | 0.0092 | 0.3332 | 810 | 1.8760 | 4.0564 | - | 0.0671 | 0.7541 | 372 |
| | Ours | **0.8861** | **1.3339** | **0.8997** | **0.0087** | **0.3269** | 906 | **1.8182** | **3.7292** | **1.8656** | **0.0598** | **0.7249** | 522 |

# 4 EXPERIMENTS

## 4.1 EXPERIMENT SETTINGS

**Datasets**. This paper evaluates the model on two public real-world crowd datasets, GC (Yi et al., 2015) and UCY (Lerner et al., 2007). These two datasets have significant differences in scene type, scale (indoor scene/outdoor scene), pedestrian density, behavior pattern, etc., which can effectively verify the generalization performance of the model in different environments. Specifically, we follow the experimental settings in PCS (Zhang et al., 2022) and SPDiff(Chen et al., 2024): select the same 5-minute trajectory data containing rich pedestrian interactions from the GC dataset for training and testing; select the same 216-second labeled trajectory data (Students003) from the UCY dataset for training and testing. We split the datasets into training and testing sets, using a training-to-testing ratio of 4:1 for the GC dataset and 3:1 for the UCY dataset.

**Implementation Details**. We train EnvSocial-Diff using Adam with a learning rate of $1e-5$ and a batch size of 32. The diffusion process uses 70 steps, and the first 25 frames of each sequence are skipped to estimate the desired walking speed. The model integrates a UNet denoiser with three conditioning modules: a 3-layer GNN for Individual–Group Interaction (IGI), pretrained ResNet-50 and BERT for Environmental Conditioning, and an LSTM encoder for up to 8 historical frames. All conditioning features are fused before predicting 2D accelerations. Additional architectural and computational details are provided in the Appendix.

**Comparison Methods**. We compare with classic physics-based and state-of-the-art data-driven and physics-informed crowd simulation methods. We choose the widely used Physics-based methods, Social Force Model (SFM) (Helbing & Molnar, 1995) and Cellular Automaton (CA) (Sarmady et al., 2010). We also compare with approaches recently published data-driven methods, including STGCNN (Mohamed et al., 2020), PECNet (Mangalam et al., 2020), MID (Gu et al., 2022), and E-$V^2$-SC (Wong et al., 2024). For physics-informed comparisons, we choose PCS (Zhang et al., 2022), NSP (Yue et al., 2022), and SPDiff (Chen et al., 2024).

**Evaluation Metrics**. We adopt the same evaluation settings and metrics as SPDiff. At the micro level, we use mean absolute error (MAE) and dynamic time warping (DTW) to assess point-wise accuracy and temporal alignment. At the macro level, we evaluate distribution similarity using optimal transport (OT) and maximum mean discrepancy (MMD). Additionally, collision count (Col) reflects how frequently predicted trajectories enter a predefined safety radius. We also introduce the final displacement error (FDE) to capture long-term prediction stability.

**Visualization results.** We present both qualitative and quantitative results on the UCY dataset in Figure 3. Panel (A) shows trajectory visualizations: in (a) and (b), the target pedestrian adjusts their path to avoid a nearby obstacle, reflecting the importance of environmental constraints; in (c), the pedestrian moves in close synchrony with familiar individuals, highlighting the effect of pairwise familiarity; in (d), the pedestrian aligns with the surrounding group flow while simultaneously avoiding oncoming pedestrians, demonstrating the need to model both group-level conformity and collision avoidance. Panel (B) further reports error curves (MAE and OT) across prediction horizons, where our method consistently maintains lower errors than SPDiff, especially in long-term predictions. These results confirm that explicitly modeling environmental cues and individual–group interactions improves both trajectory plausibility and long-horizon accuracy.

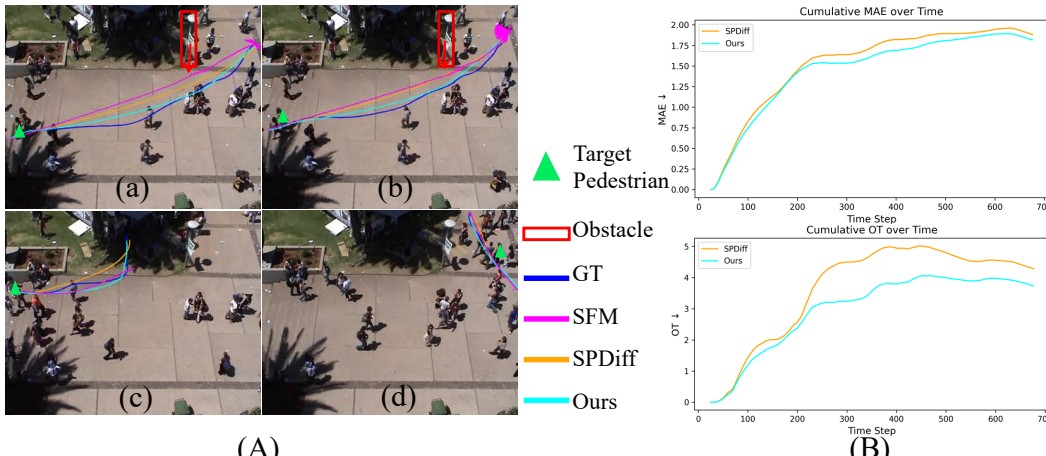

Figure 3: Comparison with baselines on UCY. (A) Predicted trajectories: our method (cyan) follows the ground truth (blue) more closely than SFM (magenta) and SPDiff (orange). (B) Error curves over time: our method consistently achieves lower MAE and OT, especially at longer horizons.

Table 3: Ablation study on structured environmental factors. The first two rows report results of the baseline SPDiff (Chen et al., 2024) and its variant extended with an explicit lighting module (SPDiff+Lighting). The lower block corresponds to our proposed EnvSocial-Diff (Ours), where obstacles, objects of interest (OOI), and lighting are progressively added. Results are reported on GC and UCY datasets across six metrics, with checkmarks indicating the included factors.

| Model | Environment | | | GC | | | | | | UCY | | | | | |
|---|---|---|---|---|---|---|---|---|---|---|---|---|---|---|---|
| | Obs | OOI | Light | MAE↓ | OT↓ | FDE↓ | MMD↓ | DTW↓ | Col↓ | MAE↓ | OT↓ | FDE↓ | MMD↓ | DTW↓ | Col↓ |
| SPDiff | ✓ | ✗ | ✗ | 0.9718 | 1.5450 | 0.9538 | 0.0100 | 0.3418 | 942 | 1.8853 | 4.2221 | 1.9000 | 0.0699 | 0.7496 | 634 |
| | ✓ | ✗ | ✓ | 0.9359 | 1.4345 | 0.9404 | 0.0099 | 0.3395 | 958 | 1.8395 | 3.8602 | 1.9463 | 0.0669 | 0.7357 | 622 |
| Ours | ✗ | ✗ | ✗ | 0.9127 | 1.3909 | 0.9246 | **0.0087** | 0.3261 | 946 | 1.8597 | 3.8945 | 1.9948 | 0.0626 | 0.7353 | 738 |
| | ✓ | ✗ | ✗ | 0.8990 | 1.3727 | 0.9162 | **0.0087** | 0.3253 | 1000 | 1.8337 | 3.8550 | 1.9987 | 0.0604 | 0.7259 | 730 |
| | ✓ | ✓ | ✗ | 0.8873 | 1.3455 | 0.9037 | **0.0087** | 0.3279 | 910 | 1.8271 | 3.8541 | 1.9671 | **0.0586** | **0.7212** | 648 |
| | ✓ | ✓ | ✓ | **0.8861** | **1.3339** | **0.8997** | **0.0087** | 0.3269 | 906 | **1.8182** | **3.7292** | **1.8656** | 0.0598 | 0.7249 | **522** |

## 4.2 EXPERIMENT RESULTS

In Table 2, we report the results of our proposed **EnvSocial-Diff** ('Ours') and comparison methods on two real-world datasets (GC and UCY). Except for E-$V^2$-SC and Ours, which are reproduced under the same experimental settings, all other results are directly cited from SPDiff (Chen et al., 2024). On the **GC** dataset, our approach achieves state-of-the-art performance on the MAE, OT, FDE, MMD, and DTW metrics. Since GC is an indoor subscene cropped from a larger environment, with limited environmental variation and relatively simple pedestrian behaviors, existing physics-informed models (e.g., PCS, SPDiff) already fit this dataset well, leading to performance saturation. Consequently, the improvements on GC are relatively limited, yet our method still consistently outperforms all comparisons across key metrics, demonstrating its stability and applicability.

On the more challenging **UCY** outdoor dataset, our method achieves relative improvements of 3.1%, 8.1%, 10.9%, and 3.9% on MAE, OT, MMD, and DTW, respectively, surpassing all comparison approaches and establishing new state-of-the-art results. The substantial gains on long-horizon metrics such as OT and MMD highlight the effectiveness of our environment factor modeling and Individual–Group Interaction mechanism in capturing complex crowd dynamics and reducing long-term prediction errors in outdoor scenarios.

## 4.3 ABLATION STUDY

**Ablations on Environmental Factors.** The ablation study on the effectiveness of structured environmental factors is presented in Table 3. The first two rows report results of SPDiff (Chen et al., 2024) and a variant (SPDiff+Lighting) that we reproduced with an additional explicit lighting module. The lower block corresponds to our proposed EnvSocial-Diff (Ours), where obstacles (Obs), objects of interest (OOI), and lighting (Light) are progressively added. As shown in the table, incorporating each factor consistently improves performance on both GC and UCY, with the full

Table 4: Ablation on the IGI module. Starting from relative motion $r_{ij}$, we incrementally add individual-level similarities ($\text{sim}^1_{ij}$: approach tendency; $\text{sim}^2_{ij}$: motion alignment) and the group-level similarity ($\text{sim}^3_i$: group conformity). Results on GC and UCY show consistent gains on most metrics, with the full configuration yielding the strongest overall performance.

| Model Variant | | | | GC | | | | | | UCY | | | | | |
|---|---|---|---|---|---|---|---|---|---|---|---|---|---|---|---|
| $r_{ij}$ | $\text{sim}^1_{ij}$ | $\text{sim}^2_{ij}$ | $\text{sim}^3_i$ | MAE↓ | OT↓ | FDE↓ | MMD↓ | DTW↓ | Col↓ | MAE↓ | OT↓ | FDE↓ | MMD↓ | DTW↓ | Col↓ |
| ✓ | ✗ | ✗ | ✗ | 0.9066 | 1.3897 | 0.9192 | 0.0093 | 0.3272 | 990 | 1.9055 | 4.0101 | 2.0525 | 0.0628 | 0.7730 | 580 |
| ✓ | ✓ | ✗ | ✗ | 0.8946 | 1.3570 | 0.9194 | **0.0086** | 0.3303 | 1000 | 1.8846 | 3.8502 | 2.0139 | 0.0588 | 0.7720 | 752 |
| ✓ | ✓ | ✓ | ✗ | 0.9208 | 1.4137 | 0.9242 | 0.0087 | 0.3404 | 1024 | 1.8725 | 3.7937 | 2.0262 | **0.0575** | 0.7761 | 614 |
| ✓ | ✓ | ✓ | ✓ | **0.8861** | **1.3339** | **0.8997** | 0.0087 | **0.3269** | **906** | **1.8182** | **3.7292** | **1.8656** | 0.0598 | **0.7249** | **522** |

model achieving the best results across most metrics (MAE, OT, FDE, MMD, DTW). This demonstrates the effectiveness of explicitly modeling structured environment cues in crowd simulation. It is noteworthy that, on the UCY dataset, adding Lighting slightly increases MMD and DTW in our model, likely due to the weaker correlation between lighting and local pedestrian dynamics in outdoor scenes. However, other key metrics (MAE, OT, FDE) continue to decrease, indicating that Lighting still contributes positively to overall prediction quality. This trend is also observed in the SPDiff baseline, where adding explicit lighting yields consistent improvements, further validating the general effectiveness of lighting as an environmental cue in trajectory prediction.

**Ablations on Similarity Terms.** The ablation study on the effectiveness of similarity terms in the Individual–Group Interaction (IGI) module is presented in Table 4. The first row corresponds to using only the relative motion descriptor $r_{ij}$, which is analogous to the interaction formulation in SPDiff (Chen et al., 2024), where social forces are conditioned purely on relative position and velocity without explicit similarity measures. The following rows progressively incorporate the three similarity terms—$\text{sim}^1_{ij}$ (approach tendency), $\text{sim}^2_{ij}$ (motion alignment), and $\text{sim}^3_i$ (group conformity). As shown in the table, adding each similarity term improves performance on both GC and UCY datasets, and the full configuration achieves the best overall results across most metrics (MAE, OT, FDE, DTW). Notably, $\text{sim}^3_i$ alone achieves a lower MMD on GC, but the absence of $\text{sim}^1_{ij}$ or $\text{sim}^2_{ij}$ weakens other metrics, confirming that group conformity alone is insufficient. These results demonstrate that modeling complementary aspects of pedestrian interactions through explicit similarity measures leads to more accurate and socially compliant trajectory forecasts. **See more experiments in the Appendix.**

## 5 CONCLUSION

This paper presents an Env–Social Physics-Informed Crowd Simulation framework that integrates environmental conditioning—including obstacles, objects of interest, and lighting—with an Individual–Group Interaction (IGI) module into diffusion-based Social Force models. By modeling these elements as physical forces and embedding them into learning architectures, our framework enables more realistic and context-aware trajectory predictions. Experiments demonstrate that incorporating environmental conditioning and the proposed IGI module significantly improves simulation accuracy, particularly in complex outdoor scenes. Our approach highlights the critical role of environmental cues in crowd motion modeling while simultaneously achieving effective social interaction modeling. Beyond trajectory simulation, future work will extend to video-level generation based on predicted trajectories, further enhancing the framework's utility for real-world crowd simulation, safety planning, and intelligent infrastructure systems.

## ACKNOWLEDGMENTS

This work was supported in parts by National Key R&D Program of China (2024YFB3908500, 2024YFB3908504), NSFC (62202312), Guangdong Basic and Applied Basic Research Foundation (2023B1515120026), Shenzhen Science and Technology Program (KQTD20210811090044003), Teaching Reform Key Program (JG2024018) and Scientific Foundation for Youth Scholars and Scientific Development Funds from Shenzhen University.

ETHICS STATEMENT

Our work focuses on modeling pedestrian dynamics for crowd simulation and trajectory prediction. The proposed EnvSocial-Diff framework is designed for research and practical applications such as urban planning, public safety analysis, and intelligent transportation systems. It does not rely on personally identifiable information; all datasets (GC and UCY) used in this study are publicly available and contain only anonymized pedestrian trajectories without facial or biometric data. Nevertheless, we acknowledge that predictive models of human motion could potentially be misused for privacy-invasive surveillance or crowd control. We encourage researchers and practitioners to employ such models responsibly, respect individual privacy, and comply with relevant data protection and ethical guidelines when deploying these systems in real-world scenarios.

REPRODUCIBILITY STATEMENT

We have made every effort to ensure that our results are reproducible. All code and configuration files will be publicly available at `https://github.com/zqyq/EnvSocial-Diff`. Our paper provides detailed descriptions of the model architecture, training procedure, experimental setup, and evaluation metrics, enabling other researchers to replicate and build upon our work.

LLM USAGE STATEMENT

Large Language Models (LLMs), such as ChatGPT, were used to assist with language polishing, grammar correction, and improving the clarity of the manuscript. All technical ideas, model designs, experiments, and analyses were conceived and executed by the authors. The LLM did not generate novel research content or influence the reported scientific results.

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

## A    IMPLEMENTATION DETAILS

**Social perception range.** The social perception range is defined as the set of nearby pedestrians that are considered when modeling individual interactions. In our framework, for each pedestrian $i$, we identify their $top_k = 6$, meaning that each pedestrian interacts with their 6 closest neighbors.

**Lighting features.** To extract lighting features, we first convert the static scene image to the HSV color space and use the V channel to represent pixel-level brightness. The image is then divided into uniform grids, and for each grid, we compute the average, maximum, and minimum light intensities. The grid size varies by dataset based on the scene's spatial scale: For the **UCY** dataset ($720 \times 576$), we use a grid size of 110 pixels, resulting in an $8 \times 6$ grid. For the **GC** dataset ($1920 \times 1060$), we use a grid size of 220 pixels, resulting in a $8 \times 4$ grid.

**Model parameters.** Our model has 58.2M parameters in total, including a ResNet50 backbone (37.8M), a lightweight BERT encoder (12.6M), and a diffusion module (7.9M). Among them, 42.6M parameters participate in the forward computation.

**Training configurations**: We use Adam (lr = 1e-5, weight decay = 1e-5) with a mild StepLR decay ($\gamma = 0.999$ every 10 epochs). The batch size is 32 and the diffusion process uses 70 steps. The invalid positions are masked as NaN. For each sequence, we skip the first 25 frames and compute each pedestrian's average velocity over these skipped frames; this value is used as the desired walking speed in the destination driving term. Training runs for 160 epochs, with each epoch taking about 69 seconds.

**Model architecture**: The model consists of three components. (1) Diffusion backbone: A UNet-based denoiser predicts accelerations, conditioned on Environmental Conditioning, Individual–Group Interaction (IGI) and Historical Trajectories. (2) IGI: A 3-layer GNN operates on a 6-nearest neighbor graph constructed at each vaild timestep, encoding relative geometry and motion cues. (3) Environmental Conditioning: Scene information is extracted using pretrained ResNet-50 (resnet50-0676ba61) and BERT (bert-base-uncased); visual and textual embeddings are concatenated to form obstacle, OOI, and global scene features. (4) The Historical Trajectories up to 8 past frames are encoded using an LSTM. All conditioning signals are projected and fused into the diffusion network, followed by a lightweight MLP that outputs 2D accelerations.

**Computational setup**: On a single NVIDIA Quadro P6000 (PyTorch 1.13.1, CUDA 11.7, FP32, batch size = 32, lr=1e-5, DDIM with 50 denoising steps), our full model (42.6M parameters) requires approximately 27 FLOPs per forward pass and 2.5–9.4GB of GPU memory, with each training epoch taking 69s. For a 651-Frame sequence, inference took 349 seconds ($\approx 0.54s$ per frame), the allocated GPU memory ranges from 1519MB to 2047MB.

## B    EVALUATION METRICS

We evaluate the quality of predicted trajectories using six standard metrics: Mean Absolute Error (MAE), Final Displacement Error (FDE), Optimal Transport (OT), Maximum Mean Discrepancy (MMD), Dynamic Time Warping (DTW), and Collision Count (COL). Below are their formal definitions.

**Mean Absolute Error (MAE).** MAE computes the average $\ell_2$ displacement error over all predicted positions. Given predicted trajectories $\{\hat{\vec{p}}_i^t\}$ and ground-truth $\{\vec{p}_i^t\}$ for $N$ pedestrians over $T$ time steps, the MAE is defined as:

$$\text{MAE} = \frac{1}{NT} \sum_{i=1}^{N} \sum_{t=1}^{T} \left\| \hat{\vec{p}}_i^t - \vec{p}_i^t \right\|_2 . \tag{21}$$

**Optimal Transport (OT).** OT measures the distributional discrepancy between predicted and ground-truth pedestrian positions using the entropy-regularized Wasserstein distance. At each time $t$, the Sinkhorn distance is computed between predicted positions $\hat{P}^t = \{\hat{\vec{p}}_1^t, \ldots, \hat{\vec{p}}_N^t\}$ and ground-truth $P^t = \{\vec{p}_1^t, \ldots, \vec{p}_N^t\}$:

$$\text{OT} = \frac{1}{T} \sum_{t=1}^{T} \mathcal{W}_\epsilon \left( \hat{P}^t, P^t \right), \tag{22}$$

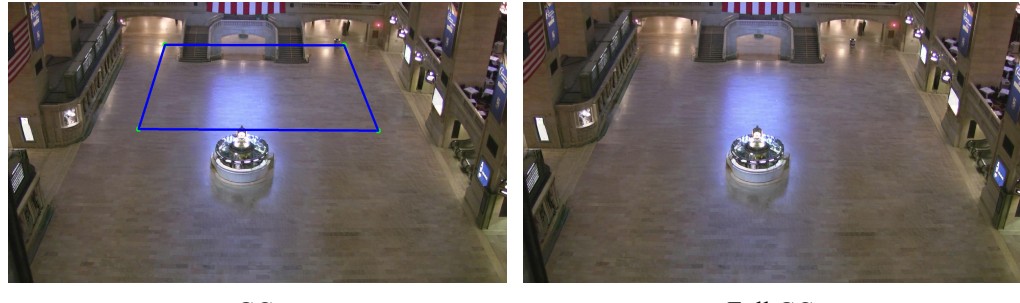

GC                                          Full GC

Figure 4: Comparison between the original GC subregion and the full GC scene. The left image highlights the cropped subarea (blue box) used in prior work, which limits spatial and interaction diversity. The right image shows the complete GC scene, covering a broader area with higher pedestrian density and environmental complexity, used in our extended evaluation.

where $\mathcal{W}_\epsilon$ denotes the Sinkhorn approximation of the Wasserstein distance with regularization coefficient $\epsilon$.

**Final Displacement Error (FDE).** FDE measures the error between the predicted and true positions at the final step. Let $T$ denote the final time step, then

$$\text{FDE} = \frac{1}{N} \sum_{i=1}^{N} \left\| \hat{\vec{p}}_i^T - \vec{p}_i^T \right\|_2 . \tag{23}$$

**Maximum Mean Discrepancy (MMD).** MMD compares the distributions of pairwise distances among pedestrians in predicted and ground-truth trajectories. Let $D_t^{\text{pred}}$ and $D_t^{\text{gt}}$ be the intra-pedestrian distance sets at time $t$, then

$$\text{MMD} = \frac{1}{T} \sum_{t=1}^{T} \text{MMD}\left( D_t^{\text{pred}}, D_t^{\text{gt}} \right), \tag{24}$$

where $\text{MMD}(\cdot, \cdot)$ denotes the kernel-based two-sample test using Gaussian kernels.

**Dynamic Time Warping (DTW).** DTW measures the similarity between two temporal sequences by computing the minimal cumulative alignment cost under temporal warping. For each pedestrian $i$, DTW distance is defined as the minimum total cost path that aligns predicted trajectory $\{\hat{\vec{p}}_i^t\}_{t=1}^T$ with the ground-truth trajectory $\{\vec{p}_i^t\}_{t=1}^T$, allowing for non-linear time alignment:

$$\text{DTW}(\hat{\vec{p}}_i, \vec{p}_i) = \min_{\pi} \sum_{(t,s) \in \pi} \left\| \hat{\vec{p}}_i^t - \vec{p}_i^s \right\|_2 , \tag{25}$$

where $\pi$ denotes a warping path satisfying boundary, continuity, and monotonicity constraints. The final DTW metric is computed by averaging over all pedestrians:

$$\text{DTW} = \frac{1}{N} \sum_{i=1}^{N} \text{DTW}(\hat{\vec{p}}_i, \vec{p}_i). \tag{26}$$

**Collision Count (COL).** COL measures the sum number of collisions among pedestrians during prediction. A collision is counted if two pedestrians $i$ and $j$ are within a certain threshold $d_{\text{thres}}$ at any time $t$:

$$\text{COL} = \sum_{t=1}^{T} \sum_{i=1}^{N} \sum_{j=i+1}^{N} \mathbb{I}\left( \left\| \hat{\vec{p}}_i^t - \hat{\vec{p}}_j^t \right\|_2 < d_{\text{thres}} \right), \tag{27}$$

where $\mathbb{I}(\cdot)$ is the indicator function.

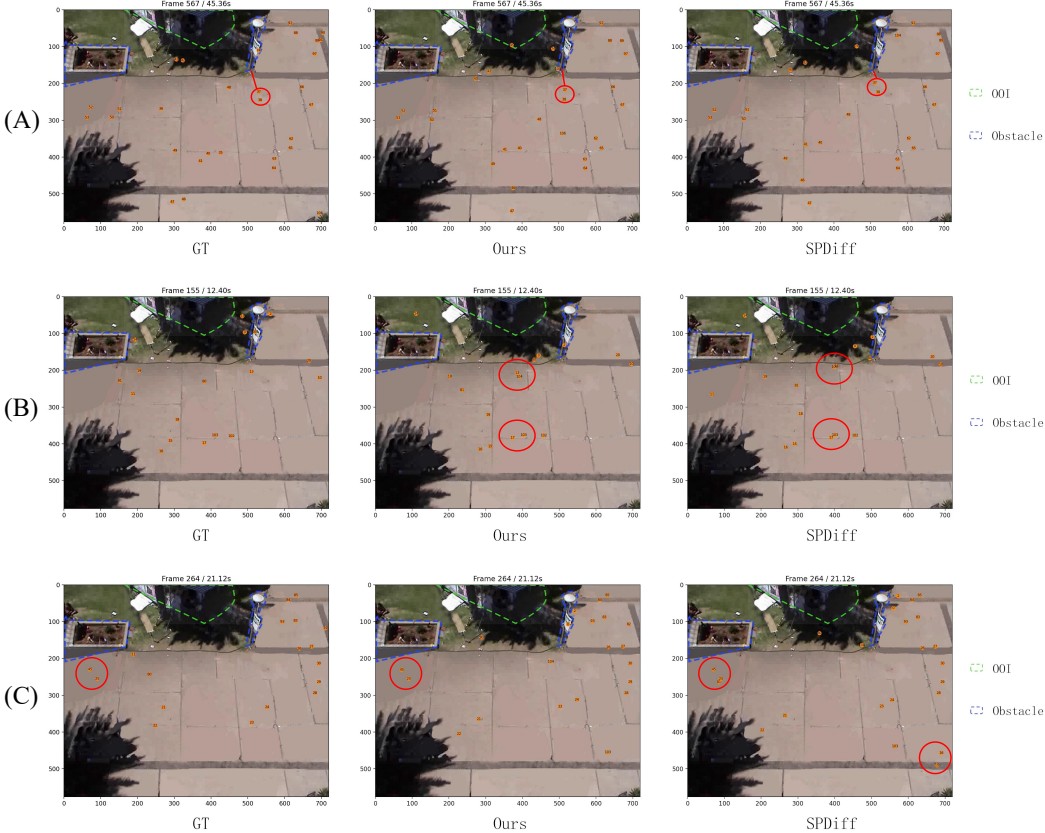

Figure 5: Qualitative comparison between the GT, Ours, and SPDiff. (A) Near obstacles, SPDiff trajectories maybe to pass much closer to obstacles, whereas both GT and our method keep a more reasonable distance. (B) In crowded regions, SPDiff produces several near-collision interactions, while our predictions remain smoother and more socially consistent. (C)

Table 5: Comparison on the full GC dataset. SPDiff uses its own social interaction module and a basic environment treatment limited to obstacle repulsion. In contrast, our method replaces the social interaction module with the proposed IGI design (✓) and additionally integrates richer environmental cues (✓), including obstacles, objects of interest, and lighting. ✗ indicates the corresponding module is not used. In the table, **Social** and **Env** are marked with '*' for SPDiff's built-in designs (social interaction and obstacle-only environment), ✓ when replaced by our modules, and ✗ when omitted.

| Dataset: Full GC | | | | | | | | |
|---|---|---|---|---|---|---|---|---|
| Method | Social | Env | MAE↓ | OT↓ | FDE↓ | MMD↓ | DTW↓ | COL↓ |
| SPDiff | * | * | 2.4624 | 4.6824 | 2.6707 | 0.0044 | 0.8873 | 1910 |
| Ours | ✓ | ✗ | 2.4478 | 4.4662 | 2.6683 | 0.0044 | 0.8836 | 1750 |
| | ✓ | ✓ | 2.3527 | 4.1334 | 2.5891 | 0.0039 | 0.8473 | 1710 |

## C  ADDITIONAL EXPERIMENTS

**Performance on Full GC scene.** To further evaluate the generalizability of our method, we conduct an additional experiment on the full GC scene. While the original GC benchmark restricts evaluation to a manually selected subregion, we apply our model to the entire scene without spatial cropping or manual filtering (see Figure 4). This setting introduces greater variability in pedestrian density, layout complexity, and environmental interactions, posing a significant challenge to prediction models.

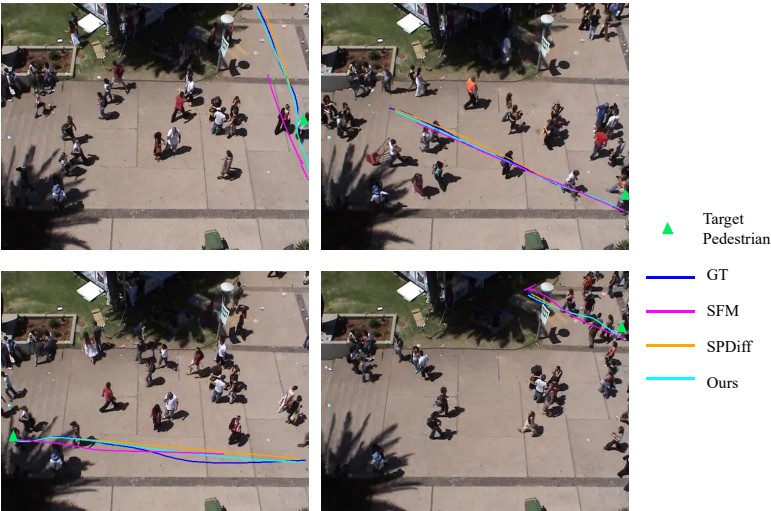

Figure 6: Additional qualitative results for multiple target pedestrians. Each subplot visualizes one scene with a selected target pedestrian (green triangle). Ground-truth future trajectory (GT) is shown in blue, while predictions from SFM (magenta), SPDiff (orange), and our method (cyan) are overlaid. Our approach consistently aligns more closely with the GT across diverse scenarios.

Table 6: Ablation on enhanced environmental features. We compare four settings: no enhancement (✗, ✗), OOI-only enhancement (✗, ✓), obstacle-only enhancement (✓, ✗), and joint enhancement with both (✓, ✓). Results on GC and UCY datasets show that the joint enhancement strategy achieves the most consistent improvements across metrics.

| Env | | GC | | | | | | UCY | | | | | |
|-----|-----|------|-----|------|------|-----|------|-----|------|-----|------|------|-----|
| Obs | OOI | MAE↓ | OT↓ | FDE↓ | MMD↓ | DTW↓ | Col↓ | MAE↓ | OT↓ | FDE↓ | MMD↓ | DTW↓ | Col↓ |
| ✗ | ✗ | 0.9320 | 1.4425 | 0.9391 | 0.0094 | 0.3362 | 926 | 1.9320 | 3.9726 | 2.0367 | 0.0609 | 0.7706 | 550 |
| ✗ | ✓ | 0.9038 | 1.3645 | 0.9164 | 0.0090 | 0.3333 | **898** | 1.9119 | **3.7120** | 2.2160 | 0.0606 | 0.7563 | 688 |
| ✓ | ✗ | 0.8960 | 1.3562 | 0.9160 | **0.0086** | 0.3277 | 962 | 1.8346 | 4.0420 | 1.9951 | 0.0634 | **0.7042** | 710 |
| ✓ | ✓ | **0.8861** | **1.3339** | **0.8997** | 0.0087 | **0.3269** | 906 | **1.8182** | 3.7292 | **1.8626** | **0.0598** | 0.7249 | **522** |

As shown in Table 5, although SPDiff uses its own social interaction module and an obstacle-only environment treatment, replacing the interaction module with our IGI design (✓) yields consistent improvements across all metrics. Furthermore, by additionally integrating richer environmental cues (✓)—including obstacles, objects of interest, and lighting—our model achieves further gains, notably reducing OT by 11.7% (4.6824 → 4.1334), MMD by 11.4% (0.0044 → 0.0039), and COL by over 10% (1910 → 1710). These results confirm the effectiveness of our IGI-based interaction modeling and demonstrate the robustness of our environment-aware framework under complex real-world conditions.

**Enhanced Features for Environmental Conditioning Modeling**. The ablation study evaluates the impact of enhanced environmental features. We start with a model without enhancement (w/o Enhance), then apply enhancement only to obstacles (Obs-Only) or only to OOI (OOI-Only), and finally apply enhancement to both (Full). As shown in Table 6, incorporating either obstacles or OOI individually improves performance, while the Full setting achieves the best results across nearly all metrics. These findings indicate that leveraging global scene context for both obstacles and OOI provides a more comprehensive modeling of environmental effects, thereby improving trajectory prediction accuracy.

Table 8: Ablation study of adding environmental factors to the SFM method on **UCY Zara1**. ✓ / ✗ indicate whether the corresponding environmental module is enabled. The results show that incorporating obstacle, lighting, and object-of-interest (OOI) cues progressively improve trajectory prediction performance in terms of MAE, MMD, and OT.

| Dataset: UCY Zara1 | | | | | |
|---|---|---|---|---|---|
| Obstacle | Lighting | OOI | MAE↓ | MMD↓ | OT↓ |
| ✗ | ✗ | ✗ | 2.5954 | 9.2648 | 1.6676 |
| ✓ | ✗ | ✗ | 1.9585 | 5.2413 | 1.1437 |
| ✓ | ✓ | ✗ | 1.8981 | 5.1045 | 1.3224 |
| ✓ | ✓ | ✓ | 1.8282 | 4.7196 | 1.0968 |

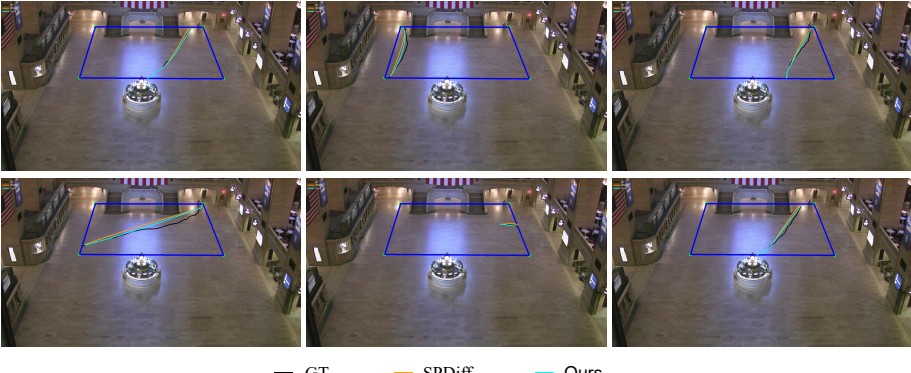

— GT    — SPDiff    — Ours

Figure 7: Qualitative comparisons on the GC dataset. Each subplot shows the predicted trajectory of a target pedestrian within the marked evaluation area (blue rectangle). Ground-truth (GT) future trajectories are depicted in black, while predictions from SPDiff and our method are shown in orange and cyan, respectively. Our approach produces more accurate and socially plausible predictions, particularly in scenarios involving sharp turns, long-range navigation, or subtle environmental conditioning.

Table 7: Ablation study of adding environmental factors to the SFM method on **UCY Students03**. ✓ / ✗ indicate whether the corresponding environmental module is enabled. Progressive addition of obstacle, lighting, and object-of-interest (OOI) cues leads to consistent improvements across all evaluation metrics.

| Dataset: UCY Students03 | | | | | | |
|---|---|---|---|---|---|---|
| obstacle | lighting | OOI | MAE ↓ | MMD ↓ | OT ↓ | COL ↓ |
| ✗ | ✗ | ✗ | 2.8943 | 7.9564 | 0.0954 | 308 |
| ✓ | ✗ | ✗ | 2.7822 | 7.3759 | 0.0899 | 266 |
| ✓ | ✓ | ✗ | 2.7324 | 7.2187 | 0.0803 | 216 |
| ✓ | ✓ | ✓ | 2.6742 | 6.7032 | 0.0708 | 216 |

**SFM + Environmental Factors.** To further assess the effectiveness of the proposed environmental modeling modules, we perform ablation studies based on the classic Social Force Model (SFM) by incrementally incorporating our three types of environmental factors: obstacles, lighting, and objects of interest (OOI). These experiments are conducted on the *UCY* dataset, focusing on two distinct scenes: *Zara1* and *Students03*.

Starting from the standard SFM as the baseline, we introduce the environmental components one by one. As shown in Tables 7 and 8, each added component consistently improves prediction accuracy. Obstacles reduce collisions and improve short-term precision; lighting enhances motion smoothness; and OOIs capture higher-level behavioral tendencies.

**Sensitivity Analyses.** In addition, we conduct a series of sensitivity analyses on three key hyperparameters: the number of cloest neighbors used for individual-level similarity computation Top k

(Table 9), the number of diffusion denoising steps (Table 10), and the spatial grid size (Table 11) for environmental encoding. The results, show that the model is overall stable under variations of these hyperparameters. For Top k, values between 2 and 8 yield comparable performance, while $k = 6$ provides the most balanced results across all metrics on both GC and UCY. Similarly, diffusion step counts from 50 to 80 exhibit only marginal fluctuations, with 70 steps offering a consistent balance between accuracy and stability. Finally, the environmental grid size demonstrates moderate influence on performance, where GC performs best around a resolution of 220 and UCY around 110, reflecting differences in scene scale. These observations indicate that the proposed model is not highly sensitive to hyperparameter choices, and the selected default settings provide a robust trade-off across datasets.

Table 9: Sensitivity analysis of the hyperparameter Top k on GC and UCY datasets. Top k controls how many nearest neighbors are used when computing individual-level similarities within the IGI module. All metrics follow the lower-is-better convention ($\downarrow$). We adopt $k = 6$ as it provides the most balanced overall performance.

| | GC Dataset | | | | | | UCY Dataset | | | | | |
|---|---|---|---|---|---|---|---|---|---|---|---|---|
| $Top\_k$ | MAE$\downarrow$ | OT$\downarrow$ | FDE$\downarrow$ | MMD$\downarrow$ | DTW$\downarrow$ | Col$\downarrow$ | MAE$\downarrow$ | OT$\downarrow$ | FDE$\downarrow$ | MMD$\downarrow$ | DTW$\downarrow$ | Col$\downarrow$ |
| 2 | 0.8918 | 1.3409 | 0.9088 | 0.0086 | 0.3273 | 902 | 1.8149 | 3.6431 | 1.9209 | 0.0596 | 0.7251 | 588 |
| 4 | 0.8896 | 1.3321 | 0.8998 | 0.0087 | 0.3284 | 914 | 1.8142 | 3.7501 | 1.9542 | 0.0601 | 0.7007 | 552 |
| 6 (Ours) | 0.8861 | 1.3339 | 0.8997 | 0.0087 | 0.3269 | 906 | 1.8182 | 3.7292 | 1.8656 | 0.0598 | 0.7249 | 522 |
| 8 | 0.8921 | 1.3599 | 0.9128 | 0.0089 | 0.3384 | 953 | 1.8154 | 3.7276 | 1.8999 | 0.0595 | 0.7159 | 476 |

Table 10: Sensitivity analysis of the diffusion step count on GC and UCY datasets. The diffusion step controls the number of denoising iterations during sampling. All metrics follow the lower-is-better convention ($\downarrow$). We adopt 70 steps as the default, as it provides a stable and well-balanced performance across all evaluation metrics.

| | GC Dataset | | | | | | UCY Dataset | | | | | |
|---|---|---|---|---|---|---|---|---|---|---|---|---|
| Step | MAE$\downarrow$ | OT$\downarrow$ | FDE$\downarrow$ | MMD$\downarrow$ | DTW$\downarrow$ | Col$\downarrow$ | MAE$\downarrow$ | OT$\downarrow$ | FDE$\downarrow$ | MMD$\downarrow$ | DTW$\downarrow$ | Col$\downarrow$ |
| 50 | 0.8870 | 1.3361 | 0.9019 | 0.0087 | 0.3269 | 984 | 1.8173 | 3.7486 | 1.8596 | 0.0616 | 0.7148 | 486 |
| 60 | 0.8860 | 1.3355 | 0.9006 | 0.0087 | 0.3273 | 912 | 1.8142 | 3.7358 | 1.8688 | 0.0603 | 0.7124 | 432 |
| 70(Ours) | 0.8861 | 1.3339 | 0.8997 | 0.0087 | 0.3269 | 906 | 1.8182 | 3.7292 | 1.8656 | 0.0598 | 0.7249 | 522 |
| 80 | 0.8862 | 1.3449 | 0.8999 | 0.0088 | 0.3285 | 908 | 1.8214 | 3.8168 | 1.8705 | 0.0612 | 0.7118 | 582 |

Table 11: Sensitivity analysis of the grid size used to construct the global scene representation on GC and UCY datasets. The grid size controls the spatial resolution of the environmental encoding, with lower-is-better metrics ($\downarrow$). We adopt a grid size of 220 for GC and 110 for UCY, as these settings yield the most balanced performance across the evaluation metrics for each dataset.

| | GC Dataset | | | | | | UCY Dataset | | | | | |
|---|---|---|---|---|---|---|---|---|---|---|---|---|
| Grid_Size | MAE$\downarrow$ | OT$\downarrow$ | FDE$\downarrow$ | MMD$\downarrow$ | DTW$\downarrow$ | Col $\downarrow$ | Grid_Size | MAE$\downarrow$ | OT$\downarrow\downarrow$ | FDE$\downarrow$ | MMD$\downarrow$ | DTW$\downarrow$ Col$\downarrow$ |
| 220(Ours) | 0.8861 | 1.3339 | 0.8997 | 0.0087 | 0.3269 | 906 | 90 | 1.8277 | 3.8858 | 1.8794 | 0.0608 | 0.7188 546 |
| 250 | 0.8842 | 1.3308 | 0.8999 | 0.0087 | 0.3275 | 914 | 100 | 1.8470 | 3.9977 | 1.9471 | 0.0629 | 0.7254 596 |
| 300 | 0.8848 | 1.3393 | 0.8986 | 0.0086 | 0.3276 | 916 | 110(Ours) | 1.8182 | 3.7292 | 1.8656 | 0.0598 | 0.7249 522 |
| 400 | 0.8840 | 1.3396 | 0.9000 | 0.0086 | 0.3276 | 918 | 120 | 1.8314 | 3.8239 | 1.9613 | 0.0600 | 0.7196 561 |

**Repulsive Force.** We implemented a simple repulsive-force variant: $a_i = a_i^{\text{ours}} + a_i^{\text{rep}}$, $a_i^{\text{rep}} = \lambda \sum_j \exp(-d_{ij}/\sigma), n_{ij}$, where $d_{ij}$ is the inter-agent distance and $n_{ij}$ is the unit vector from $j$ to $i$. As shown in Table 12, adding this repulsive-force term lowers collision counts but consistently worse MAE and OT on both GC and UCY, indicating that simple additive repulsion does not improve overall trajectory quality.

Table 12: Comparison between our method and the repulsive-force variant.

| Dataset | MAE↓ | OT↓ | FDE↓ | MMD↓ | DTW↓ | COL↓ |
|---------|------|-----|------|------|------|------|
| GC (repulsive) | 0.8970 | 1.3827 | 0.8974 | 0.0087 | 0.3370 | **894.0** |
| GC (ours) | **0.8861** | **1.3339** | 0.8997 | **0.0087** | **0.3269** | 906 |
| UCY (repulsive) | 1.8609 | 3.8766 | 1.9302 | 0.0646 | **0.7066** | **510.0** |
| UCY (ours) | **1.8182** | **3.7292** | **1.8656** | **0.0598** | 0.7249 | 522 |

**Environment-Free Diffusion Variant.** To assess the contribution of the environmental conditioning within the diffusion process, we further evaluate a simplified variant in which all environmental forces are removed from the denoiser inputs and applied only afterward in a post-processing manner. As shown in Table 13, the performance on GC changes only marginally, which is expected since GC contains stable indoor layouts with limited environmental diversity. In contrast, the gap becomes substantially larger on UCY, where open spaces and heterogeneous obstacle configurations make environmental cues more influential. These results confirm that embedding environmental information directly into the diffusion dynamics is particularly important in complex, environmentally varied scenes.

Table 13: Comparison between our full model and a simplified variant in which the environmental conditioning is removed from the diffusion inputs and applied only as forces outside the denoiser.

| Dataset | MAE↓ | OT↓ | FDE↓ | MMD↓ | DTW↓ | COL↓ |
|---------|------|-----|------|------|------|------|
| GC (variant) | 0.8862 | 1.3449 | 0.8999 | 0.0088 | 0.3284 | 908.0 |
| GC (ours) | **0.8861** | **1.3339** | **0.8997** | **0.0087** | **0.3269** | **906** |
| UCY (variant) | 2.0764 | 4.4494 | 2.1363 | 0.0730 | 0.7854 | 642.0 |
| UCY (ours) | **1.8182** | **3.7292** | **1.8656** | **0.0598** | **0.7249** | **522** |

**Automatic Annotations.** To assess the robustness of the method to noisy or automatically generated annotations, we replace all manually curated boxes with raw GroundSAM detections, which contain missing and inaccurate objects. As shown in Table 14, the performance degradation on GC is limited, consistent with the fact that GC is an indoor scene with simple, well-structured geometry that remains largely recoverable even under imperfect detections. In contrast, UCY exhibits a more noticeable drop: several key elements—such as the store façade that provides strong attraction cues—are missed by GroundSAM, reducing the effectiveness of environmental conditioning. Nevertheless, the errors remain within a reasonable range, indicating that the model retains a degree of robustness to annotation noise.

Table 14: Performance comparison when replacing manually curated annotations with raw Ground-SAM detections, which introduce missing and inaccurate boxes.

| Dataset | MAE↓ | OT↓ | FDE↓ | MMD↓ | DTW↓ | COL↓ |
|---------|------|-----|------|------|------|------|
| GC (auto) | 0.8919 | 1.3440 | 0.9059 | 0.0088 | 0.3365 | 1034.0 |
| GC (ours) | **0.8861** | **1.3339** | **0.8997** | **0.0087** | **0.3269** | **906** |
| UCY (auto) | 1.9083 | 3.9220 | 2.0423 | 0.0614 | 0.7466 | 644.0 |
| UCY (ours) | **1.8182** | **3.7292** | **1.8656** | **0.0598** | **0.7249** | **522** |

**ETH Dataset Generalizatio.** To further assess cross-dataset generalization, we evaluate our method on the ETH dataset. As shown in Table 15, our model consistently outperforms SPDiff across all metrics, demonstrating that the proposed approach generalizes well beyond the original GC and UCY datasets.

Table 15: Evaluation on the ETH dataset to assess cross-dataset generalization.

| Method | MAE↓ | OT↓ | FDE↓ | MMD↓ | DTW↓ | COL↓ |
|--------|------|-----|------|------|------|------|
| SPDiff | 0.4692 | 0.3153 | 1.7100 | 0.0886 | 0.2302 | 0.0 |
| Ours | **0.4083** | **0.2454** | **0.4639** | **0.0660** | **0.2162** | 0.0 |

**Density Conditioning Variant.** To evaluate whether lighting cues can be substituted by other perceptual representations, we replace the lighting conditioning with a global density feature constructed from a $K \times K$ density grid ($K = 16$) and a lightweight CNN encoder. As shown in Table 16, although density captures congestion levels, the substitution leads to a consistent drop across most metrics, indicating that density and lighting provide complementary rather than interchangeable cues. At the same time, the results demonstrate that our framework can accommodate alternative perceptual inputs without requiring architectural modifications.

Table 16: Replacing lighting conditioning with a global crowd-density representation constructed from a $K \times K$ density grid.

| Dataset | MAE | OT | FDE | MMD | DTW | COL |
|---------|-----|-----|-----|-----|-----|-----|
| GC (density) | 0.9132 | 1.4298 | 0.9447 | 0.0092 | 0.3407 | 1140 |
| GC (ours) | **0.8861** | **1.3339** | **0.8997** | **0.0087** | **0.3269** | **906** |
| UCY (density) | 1.8542 | 3.9204 | 1.9490 | 0.0606 | **0.7223** | 690 |
| UCY (ours) | **1.8182** | **3.7292** | **1.8656** | **0.0598** | 0.7249 | **522** |

## D  EXTRA VISUALIZATIONS

Figure 5 presents a qualitative comparison across three representative scenarios. In panel (A), trajectories generated by SPDiff tend to pass unnaturally close to obstacles, while both the ground truth and our method maintain safer margins. Panel (B) highlights behavior in dense crowds, where SPDiff produces multiple near-collision interactions, whereas our predictions remain smooth and socially coherent. Panel (C) further illustrates these differences in more complex configurations, consistently showing that our model better preserves realistic interpersonal spacing and obstacle-aware motion.

**UCY Dataset.** To further evaluate the effectiveness of our model, we present additional qualitative results on the UCY dataset, as illustrated in Figure 6. Each subplot depicts a different target pedestrian (green triangle) across various UCY scenes, with predicted trajectories from SFM (magenta), SPDiff (orange), and our method (cyan), overlaid against the ground-truth future trajectory (blue). Across diverse motion patterns and social contexts, our approach consistently produces more accurate and socially plausible predictions. Our method closely follows the ground-truth trajectories, even in challenging scenarios involving group movement, sharp turns, or interactions with nearby pedestrians. Compared to existing baselines, our model better anticipates the natural flow of pedestrian behavior and adapts more effectively to local dynamics and crowd density variations.

**GC Dataset.** We further demonstrate the robustness of our approach through qualitative comparisons on the GC dataset, as shown in Figure 7. Each subplot shows the predicted trajectory of a target pedestrian within the blue-marked evaluation area. The ground-truth (GT) trajectory is shown in black, with predictions from SPDiff and our method rendered in orange and cyan, respectively. Our model yields trajectories that better align with the GT, especially in scenarios involving long-distance navigation, abrupt direction changes, and spatial constraints along boundaries. These results underscore our model's improved capacity to reason over complex environments and nuanced spatial cues.

