# OpenReview forum: "EnvSocial-Diff: A Diffusion-Based Crowd Simulation Model with Environmental Conditioning and Individual-Group Interaction"
_ICLR.cc/2026/Conference — ICLR 2026 Poster_

### Official Review · Reviewer_MaaY · 2025-10-30

**Soundness:** 2
**Presentation:** 1
**Contribution:** 3
**Rating:** 2
**Confidence:** 4

**Summary:**

The paper proposes a diffusion-based crowd simulation model that jointly models structured environmental context and multi-level social interactions. EnvSocial-Diff introduces an environmental encoder that represents obstacles, points of interest, and lighting as conditioning signals for the denoising process. On the social side, an Interaction module models both interpersonal relations and group coherence using a graph-based design. These modules are fused with historical trajectories and a destination attraction term to guide generation toward socially compliant, context-aware motion. Experiments on GC and UCY show gains over state-of-the-art baselines.

**Strengths:**

- The paper presents a thorough ablation study that isolates the contribution of each module.

- The figures are clear and informative.

- The authors clearly mention the motivation behind their approach and highlight how their method differs from prior work.

**Weaknesses:**

- The paper lacks a notation explanation section. Introducing a concise notation summary and problem setup at the start of the methods section would significantly improve clarity.

- The methods section feels rushed and disorganized. Several terms and notations are introduced without being defined. The presentation lacks a logical progression, and makes it difficult to follow how each component connects to the overall framework.

- No sensitivity analysis is provided for key hyperparameters, and leaves it unclear how robust the model is to tuning or parameter changes.

- Implementation details are insufficient, and more information about training configurations, architectures, and computational setup would enhance reproducibility. Although the authors mention that code will be released in an anonymous repository, the link is not provided to reviewers.

**Questions:**

- How is the collision count defined? Does it refer to cases where predicted locations completely overlap, or is there a distance threshold?

- Are collisions computed only between people, or do they also include interactions with obstacles?

- For each dataset, what is the average number of people present per scene, and what are the minimum and maximum crowd sizes observed?

- The authors mention that their method produces "produce socially compliant, context-aware, and realistic trajectory predictions". However, it is unclear how these qualities are defined or quantified. What specific criteria or metrics are used to measure social compliance or realism in the results?

---

> ### Author Response · Authors · 2025-11-21
> **Response to Reviewer MaaY**
>
> Thank you for your valuable suggestions. Each question is answered as follows. We hope the response has addressed your concerns. We're glad to have further communication with you. The revision in the paper is marked in blue.
> ###  Q1. Introducing a concise notation summary and problem setup.
> In the revision, we added a concise “Notation Summary” table (L108-124), grouping the main symbols into pedestrian states, environmental features, social-interaction descriptors, and diffusion variables. We also added details on the model’s input/output (L191–197). This improves readability and helps readers follow the technical sections. We also corrected other minor notation issues for clarity.
> ### Q2. The methods section feels rushed and disorganized.
> As in Q1, we have added a new notation table to better present the notations in the paper. Besides, as questions proposed by reviewer YDkj, we have added more notation explanations for each module, such as $\mu,v$ (L 212) and  $b(\cdot)$ (L 276, L 290). We also added more details about the input and output of the model in paper (L191-197),and how the obtain $f^{sc},f^{\text{obs}}_l$ and $f^{\text{ooi}}_m$(L 259-263).
>
> Overall, the model consists of several modules, including Environment Conditioning, Individual–Group Interaction (IGI), and Historical Trajectory Encoding, which together provide the diffusion conditioning that guides the denoising process. In addition to the diffusion model itself, we further incorporate an explicit destination-attraction term to ensure goal-directed motion. Within the Environment Conditioning module, we distinguish three types of environmental factors: obstacles, objects of interest, and lighting conditions. For the IGI module, interactions are modeled at two complementary levels, namely Individual-level similarities and Group-level similarity.
> We have revised the manuscript accordingly, and the method description is now clearer.
> ### Q3. Key hyperparameters sensitivity analysis.
> In the revised Appendix (Tables 9--11), we present a sensitivity analysis of three hyperparameters—nearest-neighbor Top k, diffusion steps $K$, and lighting grid size—on both GC and UCY. Metric variations are consistently small across all settings, indicating that the model is very stable. The relative gains over baselines are also maintained, showing that EnvSocial-Diff does not rely on precise hyperparameter tuning.
>
>
> | | **GC** |  |  |  |  |  | **UCY** |  |  |  |  |  |
> |-------|-------|------|------|------|------|------|-------|------|------|------|------|------|
> |   **Top k**    | MAE↓ | OT↓ | FDE↓ | MMD↓ | DTW↓ | Col↓ | MAE↓ | OT↓ | FDE↓ | MMD↓ | DTW↓ | Col↓ |
> | **2** | 0.8918 | 1.3409 | 0.9088 | 0.0086 | 0.3273 | 902 | 1.8149 | 3.6431 | 1.9209 | 0.0596 | 0.7251 | 588 |
> | **4** | 0.8896 | 1.3321 | 0.8998 | 0.0087 | 0.3284 | 914 | 1.8142 | 3.7501 | 1.9542 | 0.0601 | 0.7007 | 552 |
> | **6 (Ours)** | 0.8861| 1.3339 |0.8997| 0.0087 | 0.3269 | 906 | 1.8182| 3.7292 | 1.8656| 0.0598 | 0.7249 | 522|
> | **8** | 0.8921 | 1.3599 | 0.9128 | 0.0089 | 0.3384 | 953 | 1.8154 | 3.7276 | 1.8999 | 0.0595 | 0.7159 | 476 |
>
> |  | **GC** |  |  |  |  |  | **UCY** |  |  |  |  |  |
> |------|--------|------|------|------|------|------|---------|------|------|------|------|------|
> |   **Step**  | MAE↓ | OT↓ | FDE↓ | MMD↓ | DTW↓ | Col↓ | MAE↓ | OT↓ | FDE↓ | MMD↓ | DTW↓ | Col↓ |
> | **50** | 0.8870 | 1.3361 | 0.9019 | 0.0087 | 0.3269 | 984 | 1.8173 | 3.7486 | 1.8596 | 0.0616 | 0.7148 | 486 |
> | **60** | 0.8860 | 1.3355 | 0.9006 | 0.0087 | 0.3273 | 912 | 1.8142 | 3.7358 | 1.8688 | 0.0603 | 0.7124 | 432 |
> | **70 (Ours)** | 0.8861| 1.3339| 0.8997 | 0.0087 | 0.3269 | 906 | 1.8182 | 3.7292 | 1.8656 | 0.0598 | 0.7249 | 522 |
> | **80** | 0.8862 | 1.3449 | 0.8999 | 0.0088 | 0.3285 | 908 | 1.8214 | 3.8168 | 1.8705 | 0.0612 | 0.7118 | 582 |
>
> | | **GC** |  | | |  |  | | **UCY** |  |  |  |  |  |
> |----------------|------|------|------|------|-------|-------|------------------|-------|-------|-------|-------|-------|-------|
> | **Grid Size** | MAE↓ | OT↓ | FDE↓ | MMD↓ | DTW↓ | Col↓ |  **Grid Size** | MAE↓ | OT↓ | FDE↓ | MMD↓ | DTW↓ | Col↓ |
> | **220 (Ours)** | 0.8861 | 1.3339 | 0.8997 | 0.0087 | 0.3269 | 906 | **90** | 1.8277 | 3.8858 | 1.8794 | 0.0608 | 0.7188 | 546 |
> | **250** | 0.8842 | 1.3308 | 0.8999 | 0.0087 | 0.3275 | 914 | **100** | 1.8470 | 3.9977 | 1.9471 | 0.0629 | 0.7254 | 596 |
> | **300** | 0.8848 | 1.3393 | 0.8986 | 0.0086 | 0.3276 | 916 | **110 (Ours)** | 1.8182 | 3.7292 | 1.8656 | 0.0598 | 0.7249 | 522 |
> | **400** | 0.8840 | 1.3396 | 0.9000 | 0.0086 | 0.3276 | 918 | **120** | 1.8314 | 3.8239 | 1.9613 | 0.0600 | 0.7196 | 561 |

---

> ### Author Response · Authors · 2025-11-26
> **Response to Reviewer MaaY**
>
> ### Q4.Implementation details are insufficient.
> **Training configurations.**
> We train the model using Adam (lr $=1\times10^{-5}$, weight decay $=1\times10^{-5}$) with StepLR ($\gamma=0.999$ every 10 epochs), batch size 32, and 70 diffusion steps. Invalid positions are masked as NaN. For each sequence, the first 25 frames are skipped and their average velocity is used as the desired walking speed. Training runs for 160 epochs, each taking about 69 seconds.
>
> **Model architecture.**
> The model has four parts: (1) A UNet denoiser predicts accelerations conditioned on all modules. (2) IGI: a 3 layer GNN over 6 nearest neighbor graphs encodes relative geometry and motion. (3) Environmental Conditioning: scene information is extracted using pretrained ResNet50 and BERT; visual and textual embeddings form obstacle, OOI, and global scene features. (4) Historical trajectories: up to 8 past frames are encoded by an LSTM. All conditioning signals are projected and fused before a final MLP outputs 2D accelerations.
>
> **Computational setup.**
> On a single Quadro P6000 (PyTorch 1.13.1, CUDA 11.7, FP32), the full model (42.6M params) requires $\sim$27 TFLOPs per forward pass and 2.5--9.4GB of GPU memory, with each training epoch taking 69s. For a 651-Frame sequence, inference took 349 seconds($\approx 0.54s$ per frame), the allocated GPU memory ranges from 1519MB to 2047MB.
>
> We have added these details in Appendix lines 719–738 and will release the code in the future.
> ### Q5. Collision count definition.
> We acknowledge the missing definition of the collision count (COL) in the appendix. Following
> SPDiff, COL is defined as the total number of pedestrian pair collisions during
> prediction, where a collision occurs whenever two agents $i$ and $j$ are within a distance
> threshold $d_{\text{thres}}$ at time $t$. We have added the definition of COL  to the appendix (L790--796) for clarity.
> ### Q6. Average, minimum, and maximum crowd sizes per scene.
> For UCY, the selected sequences average 132 pedestrians (min 107, max 150).
> For GC, the selected sequences average 206 pedestrians (min 166, max 248).
> ### Q7. Performance evaluation metrics.
> We follow the standard evaluation protocol used in prior work (SPDiff). As a supervised method, our
> predictions closely match the ground truth trajectories.
>
> We measure social compliance primarily through MMD(Maximum Mean Discrepancy) and DTW(Dynamic Time Warping). Lower MMD indicates that the spatial distribution and interaction patterns of the predicted crowd match real human behavior more closely, while lower DTW shows that the temporal evolution of each trajectory aligns better with natural pedestrian motion.
> Context awareness is evaluated via metrics that reflect whether trajectories respect scene structure. In particular, OT (Optimal Transport) captures systematic deviations that arise when predictions ignore environmental constraints such as obstacles or layout geometry, and is widely used to assess consistency with the scene context.
> Realism is assessed using MAE (mean displacement error over all predicted time steps and agents) and FDE (final-step displacement error), which measure how closely the predicted trajectories follow ground-truth paths and destinations.

---

> > ### Comment · Reviewer_MaaY · 2025-11-26
> >
> > Thank you for providing the changes. Most of my concerns are addressed.
> >
> > Regarding the dataset description, you note that the selected UCY sequences average 132 pedestrians (min 107, max 150). The UCY dataset also contains scenes with much smaller crowd sizes, such as the Zara sequences. Were these lower-density scenes entirely excluded from training and evaluation?
> >
> > The additions to the methods section have improved readability, but Table 1 is not referenced in the text. I recommend adding an explicit citation, for example between lines 191–197, to ensure that readers can easily connect the discussion to the corresponding table.

---

> ### Author Response · Authors · 2025-11-27
> **Response to Reviewer MaaY**
>
> Thank you for your valuable suggestions.  We have updated the paper according to your suggestions.
> ###  Q1. Regarding the dataset description, you note that the selected UCY sequences average 132 pedestrians (min 107, max 150). The UCY dataset also contains scenes with much smaller crowd sizes, such as the Zara sequences. Were these lower-density scenes entirely excluded from training and evaluation?
>
> Thank you for the suggestion. Following the experimental protocol of SPDiff, we adopt the same UCY data configuration and evaluate our method on Students003.
>
> To assess performance under lower-density conditions, we additionally evaluated our method on the ETH dataset (average 26 pedestrians per sequence, min 20, max 31) and the Zara sequence (average 36 pedestrians, min 25, max 48). These scenes contain substantially fewer interactions, providing a complementary test bed to evaluate robustness under sparse conditions.
> ### Zara
> | Method | MAE | OT | FDE | MMD | DTW | COL |
> |--------|------|------|------|------|------|------|
> | SPDiff | 1.8749 | 4.7763 | 5.9402 | 0.5653 | 0.7466 | 1440 |
> | Ours   | **1.1326** |**1.5298** | **1.6871** |**0.1828** |**0.5063** |**632** |
>
> ### ETH
> | Method | MAE | OT | FDE | MMD | DTW | COL |
> |--------|------|------|------|------|------|------|
> | SPDiff | 0.4692 | 0.3153 | 1.7100 | 0.0886 | 0.2302 | 0.0 |
> | Ours   | **0.4083** | **0.2454** |**0.4639** | **0.0660** | **0.2162** | 0.0 |
>
> Across both datasets, our method consistently outperforms SPDiff on all evaluation metrics, demonstrating strong robustness even in low-density scenarios.
>
> ### Q2. The additions to the methods section have improved readability, but Table 1 is not referenced in the text. I recommend adding an explicit citation, for example between lines 191–197, to ensure that readers can easily connect the discussion to the corresponding table.
>
> Thank you for the suggestion. We have added an explicit reference to Table 1 in the methods section. Specifically, we now cite Table 1 when introducing the structured environment entities $\mathcal{M}$ (Line 193), ensuring that readers can easily connect the notation in the text with the corresponding summary table.

---

> > ### Comment · Reviewer_MaaY · 2025-11-27
> >
> > Thank you for providing additional results and reflecting changes in the paper. All my concerns are addressed and I will increase my review score to accept.

---

### Official Review · Reviewer_DkCA · 2025-11-01

**Soundness:** 4
**Presentation:** 3
**Contribution:** 4
**Rating:** 8
**Confidence:** 4

**Summary:**

The paper introduces EnvSocial-Diff, a diffusion-based crowd simulation model informed by social physics and augmented with two novel modules:
(1) a structured environmental conditioning mechanism that explicitly encodes obstacles, objects of interest (OOI), and lighting conditions, and
(2) an Individual–Group Interaction (IGI) module that captures both fine-grained interpersonal dynamics and group-level conformity via graph neural networks.

The model extends the Social Physics Informed Diffusion Model (SPDiff, Chen et al., AAAI 2024) by incorporating richer environmental signals and multi-level social reasoning into the generative diffusion process.
Experiments on the GC and UCY datasets demonstrate state-of-the-art performance across multiple trajectory prediction metrics (MAE, FDE, OT, MMD, DTW, and collision count). Ablation studies confirm the contributions of environmental factors and the IGI module, while qualitative visualizations support the interpretability and realism of simulated trajectories.

**Strengths:**

+ Novel Integration: Elegant fusion of social physics and diffusion modeling with explicit environmental conditioning.

+ Interpretability: Maintains physically grounded meaning for forces and accelerations.

+ Comprehensive Evaluation: Multiple datasets, metrics, and ablations validate both performance and generalization.

+ General Applicability: Applicable to domains such as simulation, safety planning, and digital twin environments.

+ Strong Theoretical Foundation: Builds directly on the Social Force Model while extending its scope through learnable conditioning.

**Weaknesses:**

- Computational Complexity: The paper does not report training/inference times or resource comparisons versus SPDiff or data-driven baselines. This limits understanding of scalability in real-time simulation.

- Limited Dataset Diversity: Experiments rely mainly on GC and UCY datasets. These are standard but relatively small; inclusion of additional or synthetic datasets (e.g., ETH, SDD) would strengthen generalization claims.

- Lighting Factor Validation: The contribution of the lighting module is modest and potentially dataset-specific. A more detailed justification (e.g., psychophysical rationale or ablation under controlled illumination changes) would improve the argument.

- Minor Clarity Issues: Some notation (e.g., the dual use of f_{\text{light}} and \tilde{f}_{\text{light}}) could be clarified (e.g., f_{\text{light}}^{\text{raw}} and f_{\text{light}}^{\text{enc}}) for better readability.

**Questions:**

> Could the authors clarify whether the environmental encoders are trained jointly with the diffusion network or frozen (especially the ResNet and BERT backbones)?

> How does EnvSocial-Diff perform under dynamic environments (e.g., moving obstacles)?

> Have the authors considered testing the model’s capacity for long-horizon rollouts (>5s) to evaluate accumulation of social-environmental errors?

> Could the lighting conditioning be replaced or augmented by other perceptual features (e.g., crowd density maps, saliency maps)?

> Is the approach compatible with large-scale agent-based simulations (e.g., thousands of pedestrians), or does the GNN limit scalability?

---

> ### Author Response · Authors · 2025-11-21
> **Response to Reviewer DkCA**
>
> Thank you for your valuable suggestions. Each question is answered as follows. We hope the response has addressed your concerns. We're glad to have further communication with you.
> ### Q1. Computational Complexity.
> On a single NVIDIA Quadro P6000 GPU (pytorch 1.31.1, CUDA 11.7, FP32, batchsize=32, lr=1e5, DDIM with 50 denosing steps) on the UCY dataset.
> |Method|Params|TFLOPs|InfMem|InfTime(651 frames)|TrainMem|TrainTime(1 epoch)|
> |-----------------|---------|----------------------|------------------------------------|------------------------------|-----------------------------------|------------------------|
> |SPDiff| 0.23 M | 4.106| 545-882MB| 254.5s| 0.86–4.38GB|65s|
> | Ours | 42.6 M | 27 |1519-2047MB| 349s| 2.5–9.4GB| 69s|
>
> Our model has 58.2M parameters in total, including a ResNet50 backbone (37.8M), a lightweight BERT encoder (12.6M), and a diffusion module (7.9M). Among these, 42.6M parameters participate in the forward computation. Most of the total parameter count comes from the ResNet50 and BERT backbones.
> ### Q2. Include results on datasets (e.g., ETH, SDD).
> To further examine generalization, we additionally evaluate our method on the ETH dataset.
> |Method|MAE|OT|FDE| MMD| DTW|COL|
> |--------|--------|--------|--------|--------|--------|-----|
> |SPDiff |0.4692 |0.3153|1.7100|0.0886|0.2302|0.0|
> |Ours|**0.4083**|**0.2454**|**0.4639**|**0.0660**|**0.2162**|0.0|
>
> As shown in the new results, our model also outperforms SPDiff by a clear margin on ETH, demonstrating that the approach generalizes beyond the original GC and UCY datasets.
> ### Q3. Lighting Factor Validation.
> Rahm and Johansson (Assessing the pedestrian response to urban outdoor lighting: A full-scale laboratory study, 2018) provide clear psychophysical evidence for lighting's effect. They mention that “outdoor lighting is important for the walkability of a neighborhood and has been found to increase the level of walking after dark among all age groups.”
>
> Although the module’s impact is modest, it consistently improves performance on current datasets, indicating its usefulness. In future work, we will explore more complex environmental factors and consider larger scale crowd simulation datasets.
> ### Q4. Minor Clarity Issues.
> In the revision, we denote the raw lighting attribute as $f_{\text{light}}^{\text{raw}}$ and its MLP-encoded form as $f_{\text{light}}^{\text{enc}}$, which removes ambiguity and improves readability
> ### Q5. Clarify whether the environmental encoders are trained jointly or frozen?
> All environmental encoders are trained jointly with the diffusion model. ResNet-50 and BERT are initialized from pretrained weights but kept fully trainable to better align scene semantics with pedestrian dynamics, while only the lightweight projection and fusion layers are newly initialized.
> ### Q6. How does EnvSocial-Diff perform under dynamic environments?
> Extending EnvSocial-Diff to dynamic environments is straightforward, as obstacle appearance and semantics remain fixed and only their positions change. Supporting moving obstacles therefore requires only updating their coordinates at each timestep, which the Environmental Conditioning module naturally handles through relative geometry. Due to limited time and the lack of suitable datasets, we have not yet run dedicated experiments.
> ### Q7. Testing for long-horizon rollouts $>5s$?
> EnvSocial-Diff is already evaluated under long-horizon settings: GC and UCY rollouts span 54–60s (651–725 frames), far beyond the 5s. Fig3(B) also shows how MAE and OT change as the number of predicted frames increases, illustrating long-horizon error accumulation.
> ### Q8. Could lighting conditioning be replaced?
> We replaced the lighting conditioning with a global crowd density representation by building a $K\times K$ density grid, encoding it with a small CNN, and sampling features at each agent location. This provides global and local congestion cues.
> |Dataset|MAE|OT|FDE|MMD|DTW|COL|
> |---------|--------|--------|--------|--------|--------|------|
> |GC|0.9132|1.4298|0.9447|0.0092|0.3407|1140|
> |GC(ours)|**0.8861**|**1.3339**|**0.8997**|**0.0087**|**0.3269**|**906**|
> |UCY|1.8542|3.9204|1.9490|0.0606|**0.7223**|690|
> |UCY(ours)|**1.8182**|**3.7292**|**1.8656**|**0.0598**|0.7249|**522**|
>
> Although density captures congestion, replacing the lighting with density features leads to a noticeable drop across all metrics, indicating that the two cues are not interchangeable. Meanwhile, the experiment shows that the framework can accept alternative perceptual inputs without architectural changes.
> ###  Q9. Large-scale agent-based simulations, or does the GNN limit scalability?
> EnvSocial-Diff does not build a fully connected GNN; each pedestrian interacts only with its Top-k nearest neighbors. Although we have not tested scenarios with thousands of agents, the method has been evaluated on the Full-GC split (Table 5 and Fig.4), which contains around 300 pedestrians per sequence, and no scalability issues were observed.

---

### Official Review · Reviewer_ZL51 · 2025-11-01

**Soundness:** 3
**Presentation:** 3
**Contribution:** 2
**Rating:** 6
**Confidence:** 2

**Summary:**

This paper proposes EnvSocial-Diff, a diffusion-based crowd simulator that augments social-physics (SFM) with (1) structured environmental conditioning (obstacles, objects of interest, lighting, etc.) and (2) an Individual–Group Interaction (IGI) module that mixes pairwise and group-conformity signals via a GNN. On GC and UCY, the method reports consistent gains over SFM/PCS/SPDiff and other baselines across various metrics, with ablations for each factor.

**Strengths:**

- Many prior pedestrian prediction models are purely data-driven and overlook structured environmental and social factors. There are important factors in pedestrian simulation. This work tries to bridge the social-force ideas with modern generative modeling and scene modeling, which is refreshing.
    - Accurate modelling the prior scene information will be essential for future crowd simulation works.

- The decomposition into destination force + diffusion refinement is clean and easy to follow. It preserves interpretability while still benefiting from generative diversity.
- The group similarity and alignment logic, aggregated via GNN, is conceptually simple but grounded in real social-navigation dynamics. The ablations suggest it brings in meaningful improvement.

**Weaknesses:**

- The **demo video** is very difficult to interpret. This is currently the biggest presentation gap. As it stands, it is hard to tell what is happening, which agents belong to which group, or how environment cues influence behavior. Since one of the main claims is improved realism and responsiveness to context, the qualitative visualization should make these effects obvious. Overlays, legends, visual callouts, and side-by-side comparisons would help a lot.
- The framework layers several components (diffusion, semantic encoders, GNN, physics prior). While individually reasonable, it would be good to comment on compute cost and whether a simpler scene-encoder and diffusion baseline might achieve similar gains.
- GC and UCY are standard dataset in crowd simulation, but they are relatively small scenes. Since the paper emphasizes complex social and environmental structure, a brief discussion or qualitative example in a denser or more varied environment would help support the generality claims.

**Questions:**

- How sensitive is the method to the accuracy and granularity of scene annotations (objects of interest, lighting, obstacle maps)? For example, if these are noisy, incomplete, or generated automatically from imperfect scene-understanding systems, how does performance degrade?

---

> ### Author Response · Authors · 2025-11-21
> **Response to Reviewer ZL51**
>
> Thank you for your valuable suggestions. Each question is answered as follows. We hope the response has addressed your concerns. We're glad to have further communication with you. The revision in the paper is marked in blue.
> ### Q1. The demo video is difficult to interpret.
> We thank the reviewer for the suggestion. We improved the qualitative visualization by adding clear legends, obstacle/OOI overlays, and side-by-side comparisons, and we include several representative examples (See Appendix Fig.7). These updates make the scene context and the differences
> between SPDiff and our method easier to interpret.
> ### Q2. Computational cost and use a simpler scene-encoder.
> **Computational cost**:
> We already report the overall parameter count in the appendix; here we provide additional runtime statistics. On a single NVIDIA Quadro P6000 (PyTorch 1.13.1, CUDA 11.7, FP32, batch size 32, DDIM with 50 steps), the full model (42.6M forward parameters) requires about 27TFLOPs per forward pass and 2.5--9.4GB of GPU memory, with each training epoch taking 69s. For a 651-frame sequence, inference requires 349s ($\approx 0.54$s/frame), the allocated GPU memory ranges from 1519MB to 2047MB.
>
> **Simpler scene encoder**: The appendix already includes an ablation with a simplified scene encoder. In this variant, we still extract visual (ResNet-50) and textual (BERT) features for obstacles and points of interest, but we remove the scene-level fusion and directly encode each region independently. This leads to consistent degradation in performance (see Appendix Table 6), indicating that a simple per-region encoder without scene fusion cannot recover the same gains as our full scene encoder.
>
> We further evaluate a variant that obtain a simpler diffusion baseline by removing the environmental conditioning module from the diffusion inputs; the forces produced by the environment module are then applied directly to the agents outside the denoiser.
> | Dataset|MAE|OT|FDE|MMD|DTW|COL|
> |---------|--------|--------|--------|--------|--------|------|
> |GC(variant)|0.8862|1.3449|0.8999|0.0088|0.3284|908.0|
> |GC(ours)|**0.8861**|**1.3339**|**0.8997**|**0.0087**|**0.3269**|**906**|
> |UCY(variant)|2.0764|4.4494|2.1363|0.0730|0.7854|642.0|
> |UCY(ours)|**1.8182**|**3.7292**|**1.8656**|**0.0598**|**0.7249**|**522**|
>
> The results on GC are very close to those reported in the main paper. This is expected because GC is an indoor, structurally simple scene with stable layouts and fewer meaningful environmental variations. In such settings, the environmental cues play a less dominant role, so removing environmental conditioning produces only minor changes in the metrics. In contrast, as shown in the UCY experiments, scenes with open layouts and diverse obstacle distributions are more sensitive to environmental modeling, and the performance gap becomes substantially larger.
> ### Q3. Perform on a larger dataset.
> We follow the common experimental protocol established in prior work and therefore evaluate on the standard GC and UCY datasets. To further demonstrate generality in larger and denser environments, we additionally include results on the Full-GC split in the appendix (See Appendix Table 5 and Figure 4), which contains substantially more pedestrians, and we also include experiments on ETH to verify generalization beyond GC/UCY. Across all datasets, the framework remains stable and effective. We are exploring even larger and more complex datasets for future evaluation, though full-scale experiments require additional time and preprocessing.
>
> **ETH Dataset**
> | Method|MAE|OT|FDE|MMD|DTW|COL|
> |--------|--------|--------|--------|--------|--------|-----|
> | SPDiff | 0.4692 | 0.3153 | 1.7100 | 0.0886 | 0.2302 | 0.0 |
> | Ours|**0.4083** |**0.2454**|**0.4639**|**0.0660**|**0.2162**| 0.0 |
> ### Q4. Sensitivity to scene annotations accuracy.
> To evaluate sensitivity to noisy or automatically generated annotations, we replace these manual boxes with raw GroundSAM detections (without any correction), which introduces missing and inaccurate boxes. Performance drops noticeably:
> |Dataset| MAE|OT|FDE|MMD|DTW|COL|
> |------------|--------|--------|--------|--------|--------|--------|
> |GC (auto) | 0.8919|1.3440|0.9059|0.0088|0.3365|1034.0|
> |GC(ours)|**0.8861**|**1.3339**|**0.8997**|**0.0087**|**0.3269**|**906**|
> |UCY(auto)|1.9083|3.9220|2.0423|0.0614|0.7466|644.0|
> |UCY (ours)|**1.8182**|**3.7292**|**1.8656**|**0.0598**|**0.7249**|**522**|
>
> The degradation on GC is small because it is an indoor scene with simple, well-segmented structures, so the layout remains recoverable even with noisy detections. UCY, however, is an outdoor scene with richer contextual cues, and GroundSAM misses key objects such as the store façade, removing an important attraction cue. This leads to a more noticeable drop, though the errors remain within an acceptable range.

---

### Official Review · Reviewer_YDkj · 2025-11-01

**Soundness:** 3
**Presentation:** 3
**Contribution:** 2
**Rating:** 6
**Confidence:** 3

**Summary:**

The paper introduces EnvSocial-Diff, a diffusion-based model for crowd simulation. Conditioned on both environmental factors such as obstacles, objects of interest, and lighting and Individual–Group Interactions (IGI), the model aims to generate realistic walking trajectories for multiple agents.
Unlike prior approaches, EnvSocial-Diff explicitly models group conformity and leverages environmental cues beyond simple repulsive forces or binary traversability maps. Notably, it incorporates lighting conditions, which have been shown in behavioral studies to influence human motion but are largely overlooked in existing work. Trajectories are synthesized (as accelerations) using a diffusion model, conditioned jointly on the environment, IGI, and agent motion history.

**Strengths:**

(+) The proposed architecture effectively unifies and jointly models three environmental factors (obstacles, OOI, and lighting) with a social model (IGI), resulting in a powerful and more nuanced conditioning signal for the generative process, than ones used in prior works

(+) the paper shows significant design effort in the Individual-Group Interaction (IGI) module, which elegantly captures social dynamics at three distinct, complementary signals: approach tendency, motion alignment and crucially group conformity.

**Weaknesses:**

(-) the definition of several components is lacking. This includes
equation 1 doesn't expalin what m, v and mu are
(-) line 237: where is this global scene feature coming from?
(-) also the bias term as a function of the relative position between actor and obstacle isn't exaplined. why if the relative distance is larget, should the attention between actor and obstace grow?

(-) the setup should be spelled out in the beginning -- what do the authors mean by scene for example -- a single BEV image with annotated objects, obstacles, their locations and type, etc. what exactly is the output -- is it for all trajectories or just a sigle
accordingloy the dimensino of obstacle position for example should be states (2 i guess?)

(-) computational cost: i didn't see a runtime analysis and suspect that since each actor requires a full denoising scheme per waypoint the number of model activations is quite large. An analysis of NFE and wall clock time would be useful. Also memory consumption.

(-) the demo didn't really help me see the differences between the suggeted method and the baseline -- what are failrue cases in the baseline that are fixed or avoided byt he suggested model?

minor:
(-) the transpose op in equation 8 escaped

**Questions:**

(-) in figure 1 the store is marked in dotted red but i believe it should be an object of interest which should then be marked in green?

(-) how would the model handle spawning or removing of agents? I noticed that in the demo about 12 second in a pair of new nodes appears out of no where -- what is happening there?

(-) the collision count in the proposed method is reduced but i would argue stil seems quite high. What is the cause of that? could re-introducing the repulsive forces help?

---

> ### Author Response · Authors · 2025-11-21
> **Response to Reviewer YDkj**
>
> Thank you for your valuable suggestions. Each question is answered as follows.
> We hope the response has addressed your concerns. We're glad to have further communication with you. The revision in the paper is marked in blue.
> ### Q1. Explain: m, v, and mu in Eq.1; line 237 global scene feature; bias term.
> (1) In the revised version, we have updated the formulation of the destination-driving term and now provide explicit definitions for all compdonents: \[\vec{F}^{dest}_i = m_i \frac{v^{'}_i n_{i} - v_i}{\mu} \], where $v_i$ is the current velocity, v^{'}_{i} is the desired walking speed, and $n_{i}$ is the direction towards the destination. $m_i$ is a coefficient for individuals while $\mu$ is a global coefficient (Line 212).
>
> (2) The “global scene feature’’ is the scene-level environmental representation. We use GPT-5 to produce a concise scene description, encode the text with BERT and the BEV image with ResNet-50, and concatenate the resulting embeddings to obtain $f^{sc}$ (Lines 259-262).
>
> (3) The relative actor–obstacle offset is not used as a raw bias. It is first mapped by a small network $b(\cdot)$ into a learned embedding, so the resulting attention modulation is not a simple linear or distance-based function.
> ### Q2. The setup should be spelled out in the beginning.
> A scene is represented by a BEV image. We use GPT-5 to identify obstacles and OOIs and then manually verify their locations. Each obstacle/OOI has a cropped image patch, a text description, and a 2D BEV position. Lighting is extracted from the V-channel of the BEV image.
>
> The model takes as input each pedestrian’s 2D start and destination positions, along with the scene information (BEV image and text, obstacle/OOI image patches, their positions and text descriptions, and the lighting), and predicts accelerations to generate full trajectories for all pedestrians. We clarified this setup in the revision (L191–198).
> ### Q3. Computational cost.
> Our model predicts all agents jointly, so the number of function evaluations scales with frames $\times$ diffusion steps, rather than the number of actors.
> On a single NVIDIA Quadro P6000 (PyTorch 1.13.1, CUDA 11.7, FP32, batch size 32, lr=1e-5, DDIM-50), the full model (42.6M parameters) requires $\approx$27 TFLOPs per forward pass, GPU memory usage ranged from 2.5GB to 9.4GB, and each epoch required 69 s. For a 651-frame sequence, inference took 349 s ($\approx$0.54 s/frame), GPU memory usage ranged from 1519MB to 2470MB.
> ### Q4. The demo didn't show the difference between methods.
> To make the qualitative differences clearer, we have added an additional visualization Figure 7. This figure illustrates three representative cases:
>
> (A) SPDiff generates trajectories that pass unnaturally close to obstacles, while both the ground truth and our model maintain safer margins.
>
> (B) In crowded regions, SPDiff frequently produces intersecting or near-colliding paths; our multi-level social interaction modeling reduces such cases.
>
> (C) SPDiff sometimes keeps pedestrians in the scene long after they have exited in the ground truth, whereas our model produces trajectories that are more temporally aligned with pedestrian presence.
> ### Q5.The transpose op in equation 8; OOI issue in Fig. 1
> We appreciate the reviewer’s careful reading. We have fixed the escaped transpose operator in Eq. 8 and corrected Figure 1 by marking the store as an object of interest in green.
> ### Q6.  The sudden appearance of nodes in the demo.
> The sudden appearance of the two agents comes from the dataset rather than the model. These pedestrians receive valid annotations only from specific frames (e.g., one first appears at line~5992 in the raw file). During preprocessing, we interpolate trajectories only within frames with valid annotations, so an agent may appear abruptly when its first valid position becomes available.
> ### Q7. The collision count is high.
> Our evaluation covers long horizons (54--60 s), where small errors accumulate and a 0.5-radius criterion naturally yields many close interactions. Even the ground truth shows high counts (GC: 608, UCY: 174), indicating these interactions are inherent to the datasets.
> Following the reviewer’s suggestion, we tested a simple repulsive-force variant:
> $a_i = a_i^{\text{ours}} + a_i^{\text{rep}},\
> a_i^{\text{rep}} = \lambda \sum_j \exp(-d_{ij}/\sigma),n_{ij},$
> where $d_{ij}$ is the inter-agent distance and $n_{ij}$ is the unit vector from $j$ to $i$.The results are
> | Dataset| MAE|OT|FDE|MMD|DTW|COL|
> |---------|--------|--------|--------|--------|--------|-------|
> | GC(repulsive)| 0.8970|1.3827|**0.8974**|**0.0087**| 0.3370|**894.0**|
> |GC(ours)|**0.8861**|**1.3339**|0.8997|**0.0087**|**0.3269**|906|
> | UCY(repulsive) | 1.8609 | 3.8766 | 1.9302 | 0.0646 | **0.7066** |**510.0**|
> |UCY(ours)|**1.8182**|**3.7292**|**1.8656**|**0.0598**|0.7249|522|
>
> Repulsion lowers COL but clearly degrades MAE/OT, showing that simple additive repulsion does not improve overall quality.

---

### Author Response · Authors · 2025-12-03
**Final Comment for Paper #10714**

Dear Area Chair and Reviewers,

We sincerely thank you for your thoughtful feedback and valuable suggestions, which have greatly helped us improve the quality of our manuscript.

We would like to highlight that, after the rebuttal phase, all reviewers provided positive assessments of the submission. We have carefully addressed all concerns raised by the reviewers in our revised submission, including methodological clarifications, experimental details, and additional analyses and visualizations. These revisions have substantially improved the paper’s clarity, technical rigor, and contribution to the field. In particular, we would like to note that Reviewer MaaY, who initially assigned a score of 2, updated their rating to 8 after two rounds of rebuttal discussions, during which we carefully addressed concerns about methodology, experiments, and clarity. The reviewer explicitly stated that “All my concerns are addressed and I will increase my review score to accept.” This update reflects the reviewer’s final assessment following the clarifications provided. Due to the recent port-reset incident on OpenReview, however, the updated post-rebuttal score from Reviewer MaaY is not currently reflected in the displayed score.

We fully understand the constraints imposed by the current discussion policy, and we appreciate the effort and time invested by all reviewers and the Area Chair. We believe that the revised version now meets the high standards expected for publication at ICLR 2026.

Thank you once again for your time, careful consideration, and constructive engagement with our work.

Best regards,

The Authors of Paper #10714

---

### Meta-Review · Area_Chair_Q2BV · 2026-01-14

**Summary:**

The paper presents EnvSocial-Diff, a diffusion-based crowd simulation framework that generates realistic multi-agent trajectories by jointly modeling structured environmental factors and social interactions. Building upon social-physics–informed diffusion models, it introduces (i) explicit environmental conditioning that encodes obstacles, objects of interest, and lighting conditions, and (ii) an Individual–Group Interaction (IGI) module that captures both pairwise interactions and group-level conformity using graph neural networks. By synthesizing trajectories conditioned on environment cues, agent motion history, and multi-level social dynamics, EnvSocial-Diff extends prior work beyond simple repulsive forces or binary traversability.

**Reviewer Concerns:**

Across reviewers, the primary concerns center on insufficient clarity and rigor in methodology, evaluation, and presentation. Key issues include incomplete or missing definitions of core components and notations. Reviewers also consistently noted the lack of computational analysis (runtime, memory, scalability), limited dataset diversity that weakens generalization claims, and inadequate qualitative demonstrations that fail to clearly illustrate the advantages over baselines. Additional concerns involve insufficient justification or validation of certain modules, missing sensitivity/ablation analyses, and insufficient implementation details.

**Reviewer Scores:**

During the rebuttal period, the authors adequately addressed several major concerns raised by the reviewers, including the sensitivity of the proposed method to the accuracy and granularity of scene annotations. These clarifications were generally well received, leading some reviewers to raise their ratings. As a result, the overall recommendation trends toward a positive outcome.

---

### Decision · Program_Chairs · 2026-01-26

Accept (Poster)